# Tissue factor-dependent colitogenic CD4+ T cell thrombogenicity is regulated by activated protein C signalling

Gemma Leon [1], Paula A. Klavina [1], Aisling M. Rehill[1], Sarah E. J. Cooper [2], Anna Dominik[2], Shrikanth Chomanahalli Basavarajappa[3], James S. O'Donnell[1], Seamus Hussey[2], Patrick T. Walsh [3] & Roger J. S. Preston [1] ✉

Patients with inflammatory bowel disease (IBD) have an increased risk of venous thromboembolism (VTE), but the underlying mechanistic basis remains poorly defined. Here, we find that colitogenic CD4+ T cells express tissue factor (TF) and promote rapid TF-dependent plasma thrombin generation. TF+CD3+CD4+ T cells are present in both the colons of mice with experimental colitis and blood and colonic tissue from patients with IBD. Expression of genes involved in regulating coagulation, including Protein C (PC; encoded by *PROC)* and its receptor (*PROCR)*, are dysregulated in IBD patient gut biopsy tissues. Moreover, activated PC signalling reduces the procoagulant activity mediated by TF+CD4+ T cells. Our data thus identify TF-induced, colitogenic T cell-mediated thrombogenicity, and also demonstrate a new function for activated PC signalling in regulating T cell thrombo-inflammatory activity.

Inflammatory bowel disease (IBD) is a chronic inflammatory disorder of the gastrointestinal (GI) tract, commonly sub-grouped into Crohn's Disease (CD) and Ulcerative Colitis (UC)[1]. Approximately 10 million people suffer from this debilitating disease worldwide, and the incidence is rising[2]. The pathology of IBD, particularly CD, is largely mediated by aberrant T cell responses[1,3]. Despite the emergence of biologics targeting T cell pro-inflammatory activity, many patients remain unresponsive to these therapies, and many become resistant over time[4]. IBD patients are at greater risk of developing venous thromboembolism (VTE) compared to the general population. Adults with IBD exhibit a 3-fold higher risk of VTE[5], and this risk rises to 6-fold higher in paediatric IBD patients[5,6]. Strikingly, 22% of IBD patients who do develop VTE, die within 2 years of the first event[7], and 33% experience recurrent VTE within 5 years[8]. Several factors may contribute to this increased risk, including disease severity, microbial dysbiosis, pregnancy, treatment and GI surgery[9,10].

Despite the explicit association between VTE and IBD, the molecular pathogenesis of VTE in IBD patients remains poorly understood.

Plasma levels of several acute phase procoagulant and anti-fibrinolytic proteins, including von Willebrand factor, plasminogen activator inhibitor 1 and fibrinogen, are elevated during IBD 'flares'[11], whereas anticoagulant proteins such as protein C (PC), protein S and antithrombin are diminished[12]. In addition, leucocytes and platelets from IBD patients have been reported to shed increased numbers of procoagulant tissue factor (TF)+ microparticles[13,14]. Notably, antibody-mediated TF inhibition decreases disease activity in mice during dextran sodium sulfate (DSS)-induced colitis, with reduced leucocyte and platelet adhesion to colonic venules, and significantly lower rates of thrombus formation[12]. Dysregulation of the anticoagulant and anti-inflammatory PC pathway has also been implicated in IBD pathogenesis. PC^low mice develop spontaneous colitis[15], and mice largely deficient in endothelial protein C receptor (EPCR), display exacerbated disease in preclinical IBD models[16]. This has been postulated to arise due to decreased EPCR and thrombomodulin (TM) expression in the inflamed microvasculature of mice with colitis, resulting in a reduced capacity for PC activation[15,17].

[1]Irish Centre for Vascular Biology, School of Pharmacy and Biomolecular Sciences, RCSI University of Medicine and Health Sciences, Dublin, Ireland. [2]National Centre for Paediatric Gastroenterology, CHI-Crumlin, Dublin, Ireland. [3]Department of Clinical Medicine, Trinity Translational Medicine Institute, Trinity College Dublin, Dublin, Ireland. ✉e-mail: rogerpreston@rcsi.ie

Colitogenic CD4[+] T cell responses are critical for the induction and perpetuation of IBD[18]. Although largely overlooked in the context of thrombosis, recent studies have indicated a role for T cells in thrombus development and regulation. For example, a novel Treg subpopulation that produces secreted protein acidic and rich in cysteine (SPARC) was found to promote thrombolysis via recruitment of CD11c[+] monocytes with enhanced fibrinolytic activity to the thrombus[19]. Conversely, T cell-targeted immunotherapies, such as immune checkpoint inhibitors, have been reported to confer an increased risk[20] and incidence[21] of VTE in cancer patients, although the mechanisms underlying this phenomenon are currently unknown.

Here, we demonstrate a novel role for CD4[+] T cells in facilitating thrombin generation and clot formation via T cell activation-induced TF activity. We show that TF expression is upregulated in the intestinal tissue of IBD patients and mice with colitis, and we report the elevated presence of CD4[+]TF[+] T cells in the gut mucosa during colitis and in the circulation of IBD patients. Furthermore, we demonstrate that activated PC cell signalling mitigates pro-thrombotic T cell responses, independent of its canonical anticoagulant activity. These data implicate enhanced T cell-mediated thrombogenicity as a potential mediator of increased VTE risk in IBD patients and highlight a new role for activated PC anticoagulant signalling in regulating colitogenic T cell thrombo-inflammatory activity.

## Results

### The colonic tissue of IBD patients and mice with colitis is enriched with CD4[+] TF[+] T cells

To evaluate coagulation parameters in IBD patients, we examined publicly available transcriptomic data of intestinal biopsies from paediatric IBD patients in the Risk Stratification and Identification of Immunogenetic and Microbial Markers of Rapid Disease Progression in Children with Crohn's Disease (RISK) study[22]. We found that intestinal tissue *F3* expression was significantly upregulated in CD patients, with either ileal (iCD) or colonic (cCD) sites of disease, when compared to non-IBD control participants (Fig. 1a). Similarly, dysregulation in *F10*, *F2R*, *THBD*, *PROC*, *PROCR*, and *PROS1* gene expression was also observed (Supplementary Fig. 1a–e). Subsequent RNA-seq analysis of rectal biopsies from the DOCHAS paediatric IBD cohort during active disease revealed similar dysregulation in *F3* and other coagulation-associated genes in UC and CD patients compared to non-IBD patient biopsies (Fig. 1b). KEGG pathway analysis suggested dysregulation in coagulation and complement pathways in the rectal tissue of CD and UC patients compared to the control population (Supplementary Fig. 2a, b). Further in silico analysis of TF expression in human immune cells that contribute to IBD pathogenesis showed TF expression was markedly upregulated in CD4[+] αβ T cells (Supplementary Fig. 3), leading us to question whether TF expression was present on CD4[+] T cells in colonic biopsies from paediatric IBD patients. Notably, we found that CD4[+]TF[+] T cells were significantly increased in the colonic tissue of IBD patients compared to patients who presented with GI discomfort, but were later shown not to have IBD (Fig. 1c, d). Next, we isolated cells from the lamina propria of fresh colonic biopsies of treatment naïve paediatric IBD patients and measured TF expression on CD4[+] T cells by flow cytometry. TF was elevated on CD4[+] T cells isolated from both IBD and inflamed non-IBD groups, but the IBD patient cohort exhibited consistently higher TF expression (88.8% vs 38.56%, Fig. 1e–h). These studies demonstrate that CD4[+] TF[+] T cells exist and are elevated in the gut of IBD patients.

To investigate the potential role of TF in colitogenic T cells, CD4[+]CD25[low]CD45Rβ[high] T effector cells were FACs sorted from C57B6 wild-type donor mouse splenocytes and injected into the peritoneum of *Rag1[−/−]* host mice. Disease progression was measured by weight loss. Once mice had lost 20% of their original body weight, the experiment was terminated, and colons were harvested for analysis (Fig. 2a). After 4 weeks, clinical signs of disease were present in the T cell-recipient

group, as evidenced by the loss of 20% body weight (Fig. 2b). H&E histology of colonic sections demonstrated that T cell-recipient mice displayed significantly increased disease scores compared to PBS recipients ($p < 0.0001$; Fig. 2c). Notably, TF expression was significantly upregulated in the colons of T cell transfer-recipient mice compared to vehicle-treated control mice (Fig. 2d, e) and was specifically detected both in the inflamed colonic epithelium and, to a lesser extent, on monocytes (Supplemental Fig. 4). To evaluate whether infiltrating T cells were one of the cell subsets expressing TF in this tissue, we co-stained colon sections with CD3, the T cell receptor (TCR), and identified the presence of TF[+]CD3[+] T cells in the colons of mice during active colitis (Fig. 2f).

### Activated inflammatory CD4[+] T cells exhibit TF-mediated thrombogenic activity

To investigate the potential thrombogenic activity of TF[+]CD4[+] T cells, we performed a bespoke CD4[+] T cell-mediated thrombin generation assay designed so that cell-associated TF would be the sole trigger for thrombin generation (Fig. 3a). We assessed human unactivated naïve T cells (θ), TCR-activated T helper cells (Th0), colitis-associated pro-inflammatory type 1 T helper cells (Th1), T helper 17 cells (Th17) and induced regulatory T cells (iTregs) (Supplementary Fig. 5a). Interestingly, the presence of Th0 and Th1 cells promoted thrombin generation significantly more effectively than θ T cells, which had little effect (Fig. 3a–d). Specifically, θ T cells exhibited a significantly longer lag time (Fig. 3b), lower peak thrombin (Fig. 3c), and a reduced endogenous thrombin potential (ETP; Fig. 3d) compared to Th0 and Th1 cells. Th17 and iTregs also exhibited enhanced procoagulant properties (Supplementary Fig. 6). Using a plasma clotting assay, we also observed a significantly increased rate of plasma clot formation in the presence of Th0 and Th1 cells compared to naïve θ T cells (Fig. 3e). To confirm that the increased rate and size of thrombin generation arose from T cell TF activity, the assay was repeated in the presence of FVII-deficient plasma, which prevented rapid Th0 and Th1-enhanced thrombin generation, therefore showing a requirement for factor VII in the observed cell-dependent thrombin generation (Fig. 3f–i). In support of this, substitution with FXII-deficient plasma led to qualitatively similar results as when normal plasma was used (Supplementary Fig. 5b–e). Given the apparent role of activated T cell TF in enhanced thrombin generation, we assessed *F3* expression and TF surface expression in each T cell subset. *F3* expression (Fig. 3j), the percentage of cells expressing TF (Fig. 3k, l), cell surface TF expression (Fig. 3m), and FXa generation (Fig. 3n) were significantly increased on Th0 and Th1 cells compared to θ T cells. Enhanced FXa generation was also observed on Th17 and iTreg cells compared to θ T cells (Supplementary Fig. 6). These data indicate that TF expression is upregulated in activated and colitogenic inflammatory T cells, facilitating enhanced extrinsic tenase activity and thrombin generation.

### CD4[+] T cell activation promotes cell surface TF decryption

To investigate the mechanistic basis for T cell-mediated TF procoagulant activity, we evaluated molecular processes associated with enhanced TF activity, or 'decryption' in T cells. In the steady state, most immune cell surface TF is 'encrypted', and only adopts a procoagulant conformation in response to injury, inflammation, or cellular activation[23]. For example, the presence of the membrane phospholipid sphingomyelin (SM) in the outer leaflet of resting cells inhibits TF procoagulant activity[23]. However, the recruitment of acid-sphingomyelinase (ASMase) from lysosomes to the cell surface results in the hydrolysis of SM to ceramide and facilitates TF decryption (Fig. 4a)[23]. Following T cell activation, we observed significantly increased ASMase trafficking to the cell surface of Th0 and Th1 cells compared to θ T cells (Fig. 4b, c). Similarly, phosphatidylserine (PS) exposure on the outer leaflet increases TF procoagulant activity[23] (Fig. 4d). We therefore measured the binding of fluorescently labelled

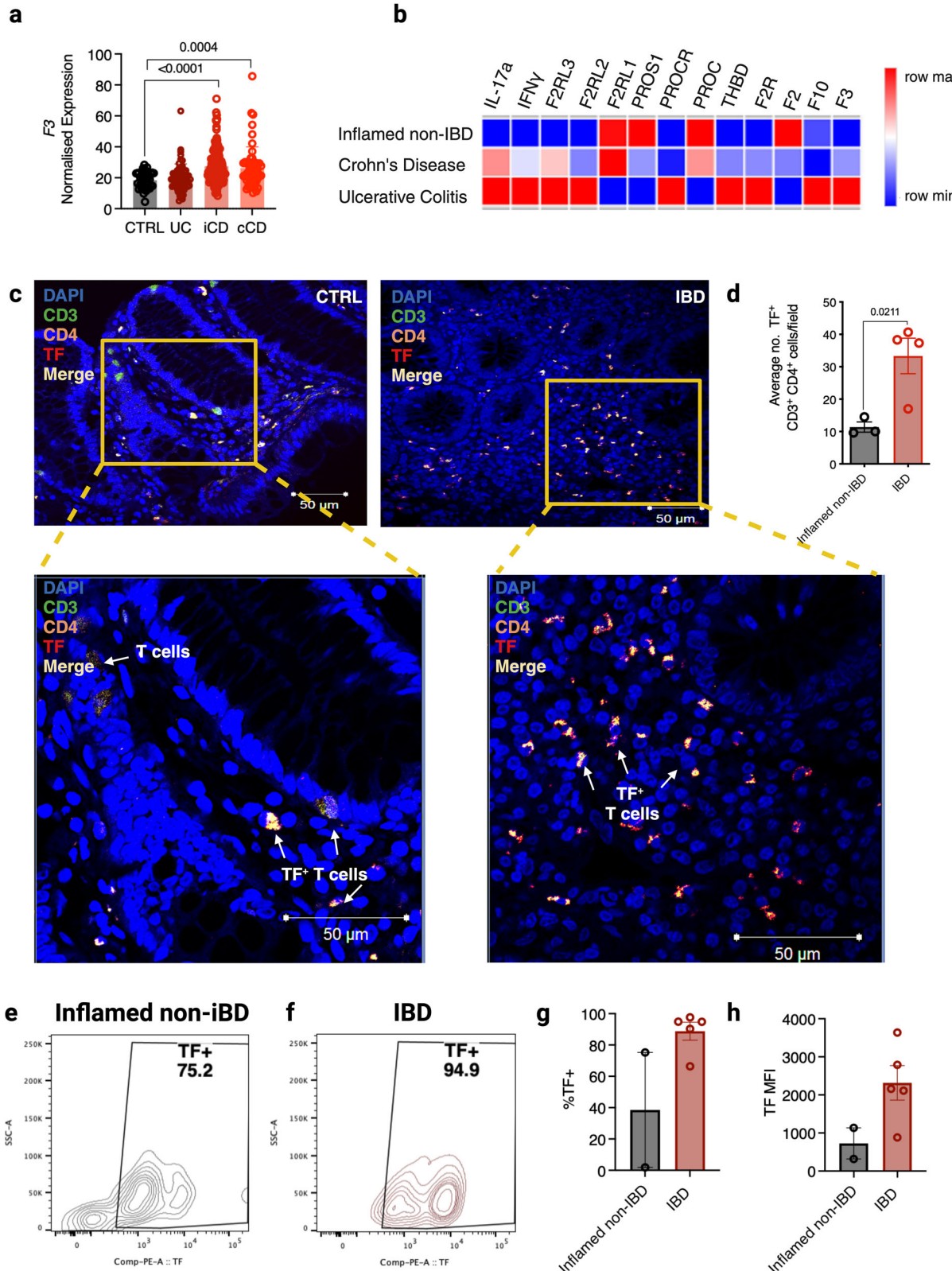

PS-binding lactadherin to the surface of CD4[+] T cells and found significantly increased lactadherin binding on Th0 and Th1 cells, compared to θ T cells (Fig. 4e, f). Translocation of the oxidoreductase enzyme protein disulfide isomerase (PDI) to the cell surface contributes to TF decryption and procoagulant activity via disulfide bond formation and thiol exchange[23] (Fig. 4g). We found that T cell activation induces a significant increase in PDI expression on the cell surface of Th0 and Th1 cells compared to θ T cells (Fig. 4h, i). Collectively, these data demonstrate that TF decryption mechanisms are responsive to T cell activation and support activation-induced TF expression and subsequent decryption as the mechanistic basis for CD4[+] T cell-mediated procoagulant activity.

**Fig. 1 | TF expression is upregulated during IBD. a** Expression of TF from RNA-seq data of ileal biopsies from Crohn's Disease (CD); illeal (iCD, $n = 162$), colonic (cCD, $n = 56$), Ulcerative Colitis (UC, $n = 62$), and query IBD healthy control cohort (CTRL, $n = 42$) patients in the RISK study (GEO ID GSE57945). **b** Expression of coagulation and inflammation-associated genes from RNA sequencing of paediatric CD ($n = 11$), UC ($n = 7$), and inflamed query non-IBD biopsies ($n = 12$). **c** TF, CD3 and CD4 co-staining and **d** the average number of TF⁺ CD3⁺ CD4⁺ T cells per field in colonic biopsies from paediatric IBD patients (IBD, $n = 4$, non-IBD, $n = 3$). **e**–**h** Lamina propria cells were isolated from colonic biopsies from paediatric IBD patients ($n = 5$)

and inflamed query non-IBD biopsies ($n = 2$). **e**–**g** The percentage of TF expressing CD3⁺ CD4⁺ TF⁺ T cells and **h** their cell surface TF expression was measured by flow cytometry and compared. Student's $t$-test (two-tailed) (**d**, **h**) or Mann–Whitney U Test (**a**, **g**) was used to determine statistical significance from **a** 322 biological donors (iCD, $n = 162$, cCD, $n = 56$, UC, $n = 62$, control non-IBD group, $n = 42$), **d** 7 biological donors ($n = 4$ IBD, $n = 3$ inflamed non-IBD), **g**, **h** 7 biological donors ($n = 5$ IBD, $n = 2$ inflamed non-IBD) and expressed as mean ± s.e.m. Source data are provided in the Source Data file.

## Peripheral CD4⁺ T cells from paediatric IBD patients exhibit TF-dependent procoagulant activity

Next, we assessed isolated CD4⁺ T cells from the peripheral blood of paediatric IBD patients and a non-IBD inflamed GI tract paediatric control population for TF expression and procoagulant activity (Fig. 5). As the non-IBD paediatric control population is not a 'healthy' comparison, we also isolated CD4⁺ T cells from the peripheral blood of healthy adult donors. We observed only a very small population of TF⁺ CD4⁺ T cells in healthy adult blood, but the peripheral TF⁺ CD4⁺ population was significantly increased in the IBD patient population (Fig. 5a–e). Next, we assessed the thrombogenic potential of CD4⁺ T cells isolated from the blood of each group. The CD4⁺ T cells from IBD patients exhibited a significantly increased capacity to initiate thrombin generation relative to T cells isolated from healthy individuals (Fig. 5f–i). Moreover, these IBD patient cohort T cells enabled 6–7 fold increased FXa generation than T cells isolated from healthy controls (Fig. 5j).

## The protein C pathway is dysregulated in IBD and regulates CD4⁺ T cell procoagulant activity

Activated PC is an important regulator of TF-mediated thrombin generation[24], IBD pathophysiology[15–17], and T cell inflammatory responses[25–30]. Therefore we next sought to investigate a potential role for activated PC anti-inflammatory signalling in regulating TF⁺CD4⁺ T cell procoagulant activity. In addition to enhanced TF expression and activity, PC (*PROC*) gene expression was significantly reduced in the colonic biopsy tissue of IBD patients compared to a non-IBD patient cohort (Figs. 1b, 6a). Activated PC bound to Th0 and Th1 CD4⁺ T cells significantly better than θ T cells (Fig. 6b, c) and, in keeping with prior studies[28], inhibited pro-inflammatory Th1 and Th17 differentiation whilst promoting the expansion of anti-inflammatory Tregs (Supplementary Fig. 6a–g, m–o). To evaluate a potential regulatory role for activated PC, we cultured each T cell subset in the presence of activated PC, washed the cells to remove residual activated PC binding, and assessed the capacity of activated PC-treated CD4⁺ cells to mediate T cell-mediated thrombin generation. Remarkably, activated PC pre-treatment of Th0 and Th1 cells also significantly decreased their capacity to facilitate thrombin generation (Fig. 6d–g, i–l). Activated PC pre-treated Th0 or Th1 cells exhibited markedly longer lag time than untreated Th0 or Th1 cells (Fig. 6e, j), significantly lower ETP (Fig. 6f, k), and significantly reduced peak thrombin (Fig. 6g, l). Similarly, activated PC pre-treatment caused a significantly decreased rate of plasma clot formation in Th0 and Th1 cells (Fig. 6h, m). To confirm this activated PC-mediated anticoagulant effect was mediated by direct reduction of TF activity, we measured FXa generation on the surface of each T cell subset in response to activated PC pre-treatment. We observed a significant reduction in FXa generation on activated PC-treated Th0 and Th1 cells, compared to untreated T cells (Fig. 6n, p). To explore this further, we analysed TF expression in activated PC pre-treated CD4⁺ T cells. We found that TF (*F3*) expression was significantly reduced in both Th0 and Th1 cells treated with activated PC (Fig. 6o, q) and that these cells also displayed significantly lower levels of PDI on their surface (Fig. 6r–u). To determine whether PDI inhibition would result in diminished TF-mediated procoagulant activity, we tested the effect of rutin, a well-characterised inhibitor of PDI activity[31], in T cell-

dependent thrombin generation analysis. Interestingly, rutin-mediated PDI inhibition on CD4⁺ T cells caused a significant reduction in ETP in both Th0 and Th1 cells (Fig. 6v–y). Notably, this inhibitory effect did not restore thrombin generation to that of naïve unstimulated T cells, suggesting PDI inhibition alone is not sufficient to prevent TF-dependent procoagulant activity on Th0 or Th1 cells. To address the potential role of EPCR and PARs in this activity, we utilised activated PC variants incapable of either EPCR binding (activated PC^ΔGLA) or PAR activation (activated PC^DEGR). Interestingly, the loss of PAR proteolysis did not significantly affect activated PC restriction of T cell-mediated procoagulant activity in Th0 or Th1 cells (Supplementary Fig. 7). In contrast, loss of EPCR binding did cause a significant increase in some thrombin generation parameters, although the effect varied between Th0 and Th1 cells. These results indicate that activated PC may utilise EPCR, but not PAR activation, to reduce T cell thrombogenicity.

## Activated PC inhibits procoagulant activity of peripheral CD4⁺ T cells from IBD patients

Finally, to ascertain whether thrombogenic T cells isolated from the peripheral blood of IBD patients were sensitive to activated PC-dependent suppression of TF procoagulant activity, isolated T cells were incubated with activated PC before the cells were washed and assessed (Fig. 7). Activated PC treatment significantly extended lag time and reduced peak and total thrombin generation (Fig. 7a–d). Moreover, FXa generation mediated by peripheral CD4⁺ T cells from IBD patients was also reduced by activated PC pre-treatment (Fig. 7e). Taken together, these data demonstrate a novel non-canonical mechanism of activated PC anti-immunothrombotic activity, namely the direct inhibition of TF-mediated CD4⁺ T cell thrombo-inflammatory activity.

## Discussion

IBD patients exhibit a significantly increased risk of VTE compared to the general population[5,6], yet the molecular pathways underpinning the pathogenesis of VTE in IBD remain poorly understood. In this study, we report the presence of TF⁺ CD4⁺ T cells in the inflamed mucosa of paediatric IBD patients and colitogenic mice, uncover the thrombogenic phenotype of activated CD4⁺ T cells, and demonstrate a new role for activated PC signalling in mitigating CD4⁺ T cell thrombo-inflammatory activity.

Several studies have highlighted coagulation dysregulation in both IBD patients and preclinical models of IBD[12–14,26,32]. This disruption is particularly evident in models of innate immune-mediated colitis, characterised by enhanced procoagulant TF activity, microparticle release and diminished PC pathway activity in the gut epithelia[12,15,17]. This dysregulation was confirmed upon transcriptomic analysis of IBD patient intestinal biopsy tissue, which revealed heightened TF expression in IBD patients compared to non-IBD patient control tissue. However, this bulk analysis did not provide insight into the breadth of cellular sources of increased TF expression in IBD patients. Subsequent in silico analysis of innate and adaptive immune cell TF expression, however, suggested that adaptive immune CD4⁺ T cells may represent a surprisingly rich source of TF. CD4⁺ T cell TF expression is greater than that typically observed on more common cellular TF sources,

such as activated monocytes or neutrophils[33,34]. As IBD pathophysiology is largely driven by T cell dysfunction[18], we examined whether CD4+ T cells may represent a novel source of thrombogenic activity in IBD patients. We found that IBD patients exhibited significantly higher numbers of TF+CD4+ T cells in their colons and periphery compared to non-IBD individuals. Furthermore, we observed increased TF

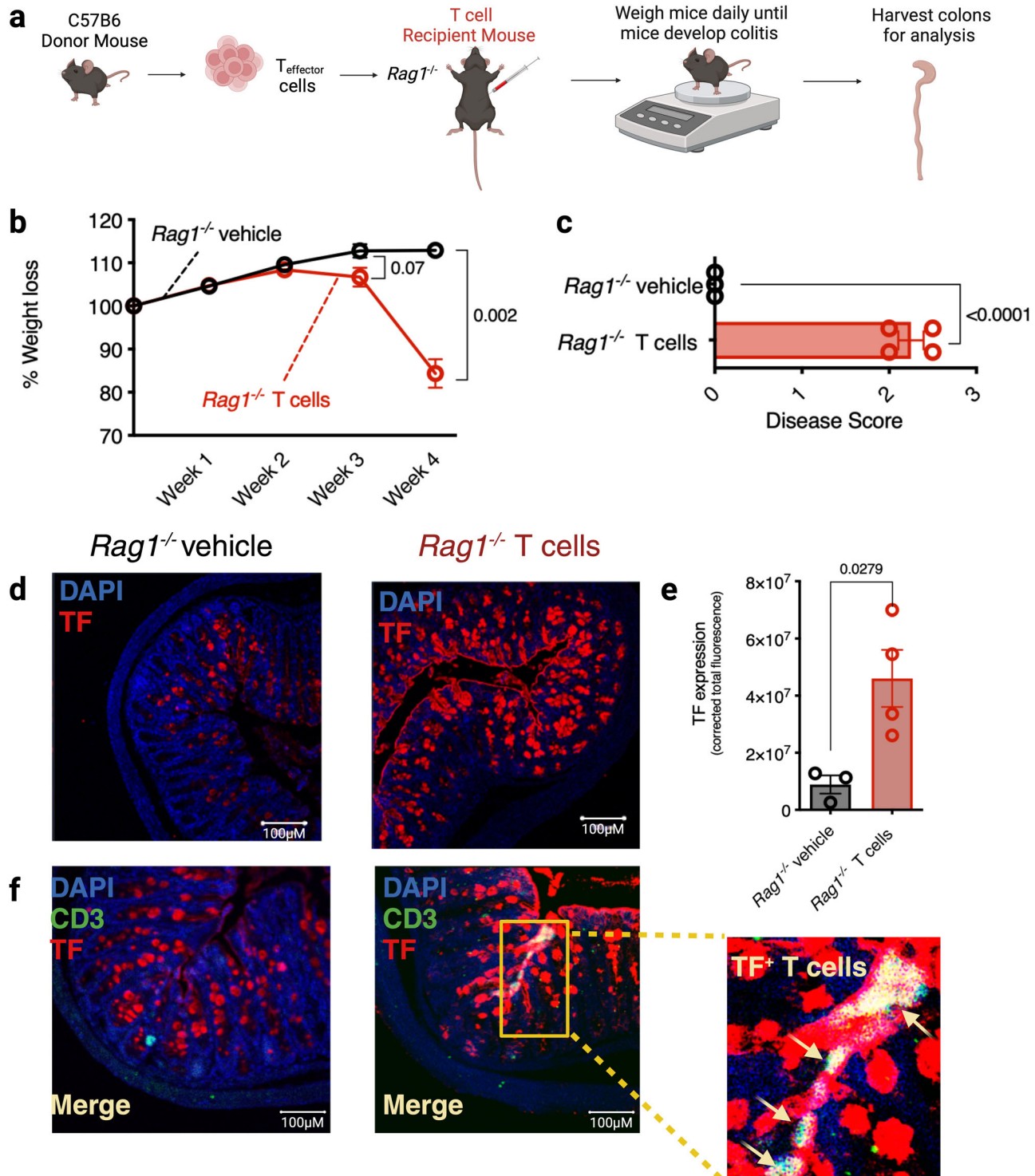

**Fig. 2 | TF expression is enhanced in the colon and on infiltrating T$_{effector}$ cells during T cell transfer-induced colitis. a** Schematic diagram of the T cell transfer model of colitis. CD4+ T effector cells were isolated from donor C57BL/6 wild-type (WT) mice and i.p. injected into host *Rag1*$^{-/-}$ mice (*Rag1*$^{-/-}$ T cells). *Rag1*$^{-/-}$ mice i.p. injected with PBS were used as a control (*Rag1*$^{-/-}$ vehicle). **b** Colitis developed over a 4-week period, and disease progression was measured by % weight loss compared to original weight. **c** This was confirmed by subsequent colon histology analysis. **d** TF staining and **e** Corrected total fluorescence (CTF) of mice colons following T cell transfer-induced colitis. **f** TF and CD3 co-staining of mouse colons following T cell transfer-induced colitis (reproduced *n* = 4 *Rag1*$^{-/-}$ T cells and *n* = 3 *Rag1*$^{-/-}$ vehicle). 2-way ANOVA (**b**) or two-tailed Student's *t*-test (**c**, **e**) was used to determine statistical significance from 3–4 biological replicates, *n* = 4 male *Rag1*$^{-/-}$ T cells and *n* = 3 male *Rag1*$^{-/-}$ vehicle, expressed as mean ± s.e.m. Scale bars: 100 μm (**d**, **f**). Source data are provided in the Source Data file. Created in BioRender. Preston, R. (2025) https://BioRender.com/o91u731.

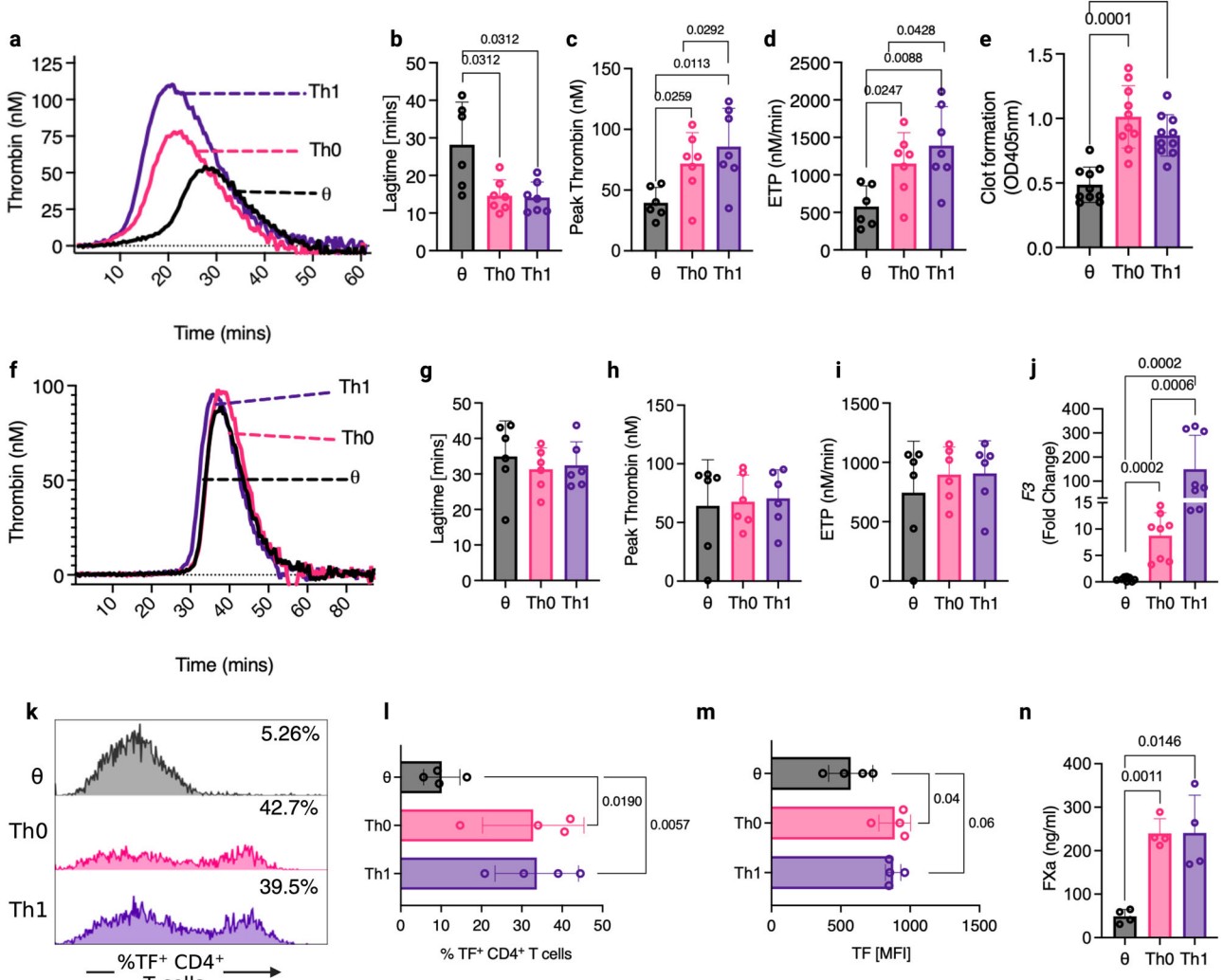

**Fig. 3 | T cell activation enhances TF-mediated CD4⁺ T cell thrombogenicity.** CD4⁺ T cells were isolated from donor human blood and plated at a density of 0.8 × 10⁶/ml with IL-2 for unactivated conditions (θ). Plated cells were activated by anti-CD3/anti-CD28 activation beads and stimulated with IL-2 for Th0 conditions, and differentiation cytokines (αIL-4 + IL-12) were added to skew cells to a Th1 lineage. **a–d** Following 5 days in culture, θ, Th0 and Th1 cells were washed with EDTA-containing PBS, and their ability to initiate thrombin generation was analysed by calibrated automated thrombinography in normal pooled platelet-poor plasma. **b** Lagtime, **c** peak thrombin levels and **d** endogenous thrombin potential (ETP) were measured and compared between θ, Th0 and Th1 cells. **e** The rate of clot formation was measured in θ, Th0 and Th1 cells. **f–i** θ, Th0 and Th1 cell-mediated thrombin generation was analysed by calibrated automated thrombinography using FVII-deficient platelet-poor plasma. **g** Lagtime, **h** peak thrombin levels and **i** ETP were measured and compared between θ, Th0 and Th1 cells. **j** *F3* gene expression, **k, l** the percentage of cells expressing TF, **m** cell surface TF expression and **n** T cell-dependent FXa generation was measured in θ, Th0 and Th1 cells. Student's paired *t*-test (two-tailed) (**c–e**, **g–i**, **l–n**), Wilcoxon test (two-tailed) (**b**), or Mann–Whitney U Test (two-tailed) (**j**) was used to determine statistical significance. Data is expressed as mean ± s.d. (**b–e**, **g–i**, **l–n**) for 7 (**a–d**), 10 (**e**), 6 (**g–i**), 8 (**j**) and 4 (**l–n**) biological donors/group. Source data are provided in the Source Data file.

expression on infiltrating immune cells and CD4⁺ T cells in the colons of mice in a T cell-mediated model of colitis. Consequently, TF⁺ CD4⁺ T cells exist in the gut and are significantly increased during active disease.

Cell surface TF is normally expressed in an 'encrypted' state, with limited procoagulant activity[23]. Several mechanisms that regulate TF adoption of a procoagulant, 'decrypted' state have been proposed. These steps typically either alter TF structural conformation (PDI-mediated disulfide bond rearrangement)[23] or adjust the phospholipid microenvironment around TF to facilitate optimal tenase complex activity (PS and SM rearrangement)[23]. We observed that key mediators of TF decryption, namely ASMase membrane translocation, PDI surface expression and PS exposure, are activated in Th0 and Th1 cells, but largely absent in naïve T cells. Although stimuli for TF decryption in innate immune cells, such as TLR activation, cytokine stimulation or toxic stressors[33], are unlikely to be shared with T cells, downstream

signalling pathways leading to TF decryption are likely to overlap. For instance, activation of p38 mitogen-activated protein kinase (p38 MAPK)-dependent signalling in monocytes promotes TF decryption that is largely mediated by PS exposure[35]. Similarly, TCR engagement also promotes p38 MAPK signalling in CD4⁺ T cells[36], which may also contribute to TF decryption. Interestingly, the expression of these decryption pathways has been implicated in several other aspects of T cell-mediated IBD pathophysiology. For example, ASMase expression in T cells enhances proliferation and survival following TCR activation[37], specifically via co-stimulatory CD28 receptor ligation[38]. In addition, ASMase-deficient mice exhibit significantly higher numbers of splenic Tregs than wild-type mice[39]. Notably, pharmacological ASMase inhibition in DSS-induced colitis significantly reduces disease severity[40,41]. PS exposure on the T cell surface has been shown to bind T cell immunoglobulin and mucin domain containing-3 (TIM-3) and act as a co-receptor for T cell activation[42]. Furthermore, PS externalised by

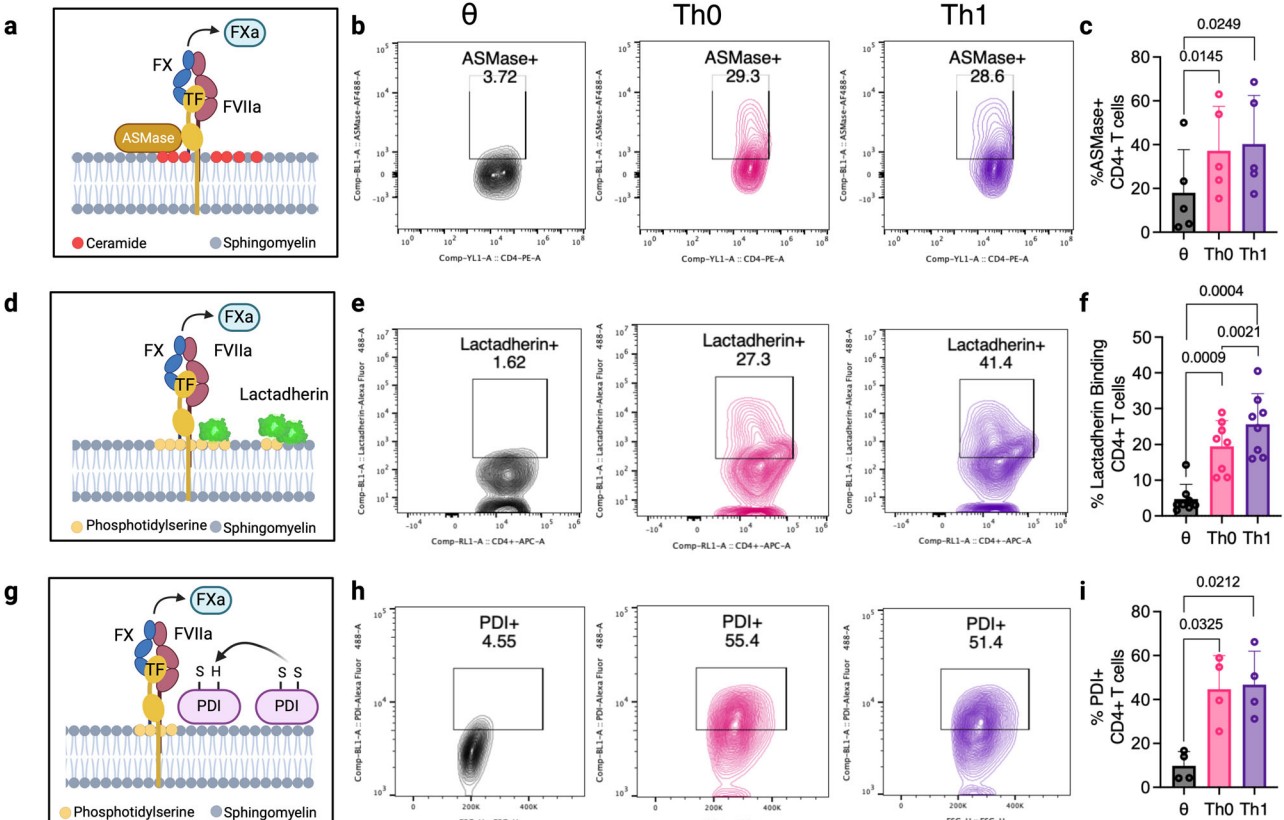

**Fig. 4 | CD4+ T cell activation upregulates markers of TF decryption.** CD4+ T cells were isolated from donor human blood and skewed to θ (unactivated cells), Th0, and Th1 cell subtypes as described previously. **a** ASMase translocation to the cell surface, **d** PS exposure on the outer membrane leaflet, and **g** PDI recruitment to the cell surface were assessed. TF decryption pathways in θ, Th0 and Th1 were analysed by **b, c** ASMase cell surface expression, **e, f** fluorescently labelled lactadherin binding to exposed cell surface PS, and **h, i** cell surface PDI expression. Student's paired *t*-test (two-tailed) (**c, f, i**) was used to determine statistical significance. Data is expressed as mean ± s.d. (**c, f, i**) for 5 (**c**), 8 (**f**), and 4 (**i**) biological donors/group. Source data are provided in the Source Data file. Created in BioRender. Preston, R. (2025) https://BioRender.com/x11c079.

antigen-stimulated CD8 T cells is linked to T cell activation, with increased expression of CD69 and levels of IFN-γ, IL-2, and TNF-α reported[43]. In trinitrobenzene sulfonic acid (TNBS)-induced colitis, PS inhibition by annexin A5 alleviated disease via reduced inflammatory cell infiltration due to inhibition of endothelial cell activation[44]. Furthermore, pups reared on artificial milk with the addition of lactadherin, the PS binding and blocking glycoprotein, displayed increased levels of tolerogenic CD3+CD4+CD25+T cells in their Payer's patches compared to pups reared on artificial milk alone[45]. There are also several reports indicating a role for PDI in regulating T cell responses in cancer and infection. PDIA3-specific T cell clones were found in colorectal cancer (CRC) patients and displayed aberrant immunity in models of malignant melanoma[46]. Furthermore, recent reports have suggested an important role for PDI in eliciting enhanced anti-parasitic responses in T cells. Immunisation with *Leishmania donovani* PDI (*Ld*-PDI) followed by challenge with the *L. donovani* resulted in enhanced CD4+ Th1 and Th17, and CD8+ T cell immunoreactivity via MAPK-pathway signalling in Balb/c mice[47]. Similarly, in studies of *Toxoplasma gondii* infection, Balb/c mice immunised with recombinant *T. gondii* PDI (*rTg*-PDI) showed enhanced Th1 immunity and reduced levels of parasitic infection following challenge with *T. gondii*[48]. Although our study used ex vivo-generated Th0 and Th1 cells to assess T cell pro-coagulant activity and TF decryption, the specific inflammatory and cellular determinants that drive activation of T cell thrombogenicity in vivo remain unknown and represent an important avenue for further investigation.

Activated PC is best characterised as a plasma anticoagulant that degrades activated cofactors factor V (FVa) and factor VIII (FVIIIa) to limit thrombin generation and subsequent fibrin deposition[24]. Like other coagulation proteases, activated PC mediates receptor-mediated cell signalling that is predominantly anti-inflammatory[49]. Activated PC signalling is typically mediated by activation of protease-activated receptors (PARs), in particular PAR1 and PAR3. PAR signalling by activated PC typically requires activated PC co-localisation via binding to a co-receptor, most commonly EPCR[50]. Notably, disruption of activated PC generation or activity has been shown to contribute to T cell-mediated inflammatory diseases[28–30]. Activated PC inhibits pro-inflammatory Th17 and Th1 activity and promotes the expansion of tolerogenic Tregs[25–28,51]. Activated PC administration ameliorates graft vs host disease (GvHD) via anti-inflammatory PAR1 signalling on T cells[51]. Furthermore, pre-incubation of donor T cells with activated PC ex vivo inhibits allo-genic T cell expansion and increases the pool of Tregs before trans-plantation via PAR2 and PAR3 signalling[28]. Activated PC also promotes altered T cell metabolism, resulting in *Foxp3* promoter demethylation[30]. EPCR-deficient (*PROCR*low) mice (that generate less activated PC and have limited capacity to facilitate EPCR-dependent activated PC signalling) exhibit exacerbated disease in experimental autoimmune encephalomyelitis, via enhanced pro-inflammatory Th17 cell generation[29]. Similarly, in preclinical models of atopic dermatitis and arthritis, activated PC administration reduces Th1 and Th17 cell populations via PAR1 and PAR2 signalling[25–27]. Our data suggest that, in addition to these potent anti-inflammatory activities, diminished activated PC activity may contribute to aberrant mucosal haemostasis and impair local regulation of T cell-mediated thrombogenicity.

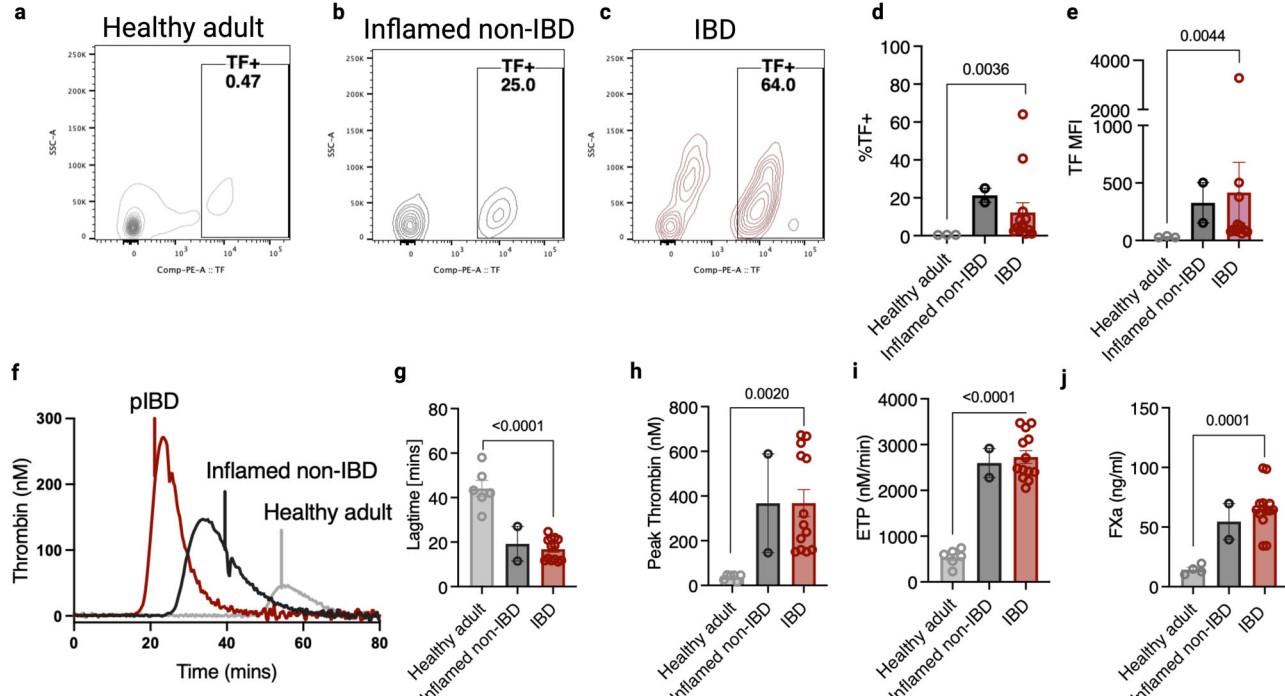

**Fig. 5 | Peripheral CD4+ T cells from IBD patients exhibit TF-mediated thrombogenicity.** CD4+ T cells were isolated from donor adult human peripheral blood, inflamed non-IBD paediatric peripheral blood and paediatric IBD peripheral blood were plated at a density of 1 × 10⁶/ml and incubated for 24–48 h at 37 °C in AIM-V media supplemented with immune replacement serum. **a–d** The percentage of cells expressing TF and **e** their cell surface TF expression was measured by flow cytometry and compared between groups. **f–j** Plated cells were washed in EDTA-containing PBS, and their ability to initiate thrombin generation was analysed by calibrated automated thrombinography in FXII-deficient plasma. **g** Lagtime, **h** peak thrombin levels and **i** ETP were measured. Furthermore, the ability of isolated T cells to facilitate FXa generation in which the T cells were the sole source of TF, was measured (**j**). Mann–Whitney U Test (two-tailed) (**d, e, g**) or Student's *t*-test (two-tailed) (**j, h, i**) was used to determine statistical significance. Data is expressed as mean ± s.e.m. (**d, e, g–j**) for **d, e, j** 17 biological donors (*n* = 12 IBD, *n* = 2 inflamed non-IBD, *N* = 3 healthy adult), **g–i** 21 biological donors (*n* = 13 IBD, *n* = 2 inflamed non-IBD, *N* = 6 healthy adult) and **j** 18 biological donors (*n* = 12 IBD, *n* = 2 inflamed non-IBD, *N* = 4 healthy adult). Source data are provided in the Source Data file.

In conclusion, we show for the first time that activated PC binding to the surface of CD4+ T cells and subsequent signalling activity directly regulates CD4+ T cell-expressed TF procoagulant activity. The ability of activated PC to limit disease-associated features of IBD in preclinical colitis models has led to its consideration as a therapy in treating IBD[15,17]. However, the potent anticoagulant activity of activated PC and its association with increased bleeding risk in patients represent a significant obstacle to its successful clinical implementation. Our study, however, suggests that recombinant 'non-anticoagulant' activated PC variants[34,52–54] that are unable to degrade FVa and FVIIIa effectively may still selectively regulate T cell-specific immunothrombotic activity, without impacting canonical activated PC anticoagulant substrates. Consequently, these findings highlight a potential therapeutic role for activated PC targeting CD4+ T cell thrombo-inflammatory activity in IBD and other T cell-mediated disease contexts.

## Methods
### Study participants
All human samples were obtained with written informed consent/assent from the legal guardians of paediatric IBD patients and control participants recruited in the Determinants and Outcomes of CHildren and Adolescents with IBD Study (DOCHAS) at the gastroenterology unit at Children's Health Ireland (CHI), Crumlin (Dublin, Ireland). All participants were aged from 0 to 17 years and underwent diagnostic evaluation according to international paediatric standards (Porto criteria) and rigorously phenotyped using the paediatric-specific Paris classification of IBD. Rectal and colonic biopsies were obtained from patients enrolled in the study. Patients initially enrolled with suspected IBD but subsequently not diagnosed with disease comprise the control population. All experiments using these tissues were performed under approval from the Children's Health Ireland Crumlin Institutional Research Ethics Committee (GEN/193/11) and included 45 participants (CD, *n* = 18; UC, *n* = 11; IBD undefined, *n* = 1, Ctrl, *n* = 14). Patient information can be found in Supplementary Table 2.

### Isolation and culture of CD4+ T cells
Anonymised healthy donor buffy coats were obtained from the Irish Blood Transfusion Service, St. James' Hospital, Dublin, Ireland. Paediatric IBD and non-IBD control blood samples were obtained from the DOCHAS study, National Centre for Paediatric Gastroenterology, CHI-Crumlin, Dublin, Ireland. PBMCs were isolated using Lymphoprep density gradient centrifugation (#18060, STEMCELL Technologies), and negative magnetic selection was then used to purify CD4+ T cells (CD4 T cell isolation Kit, human, #130-096-533, Miltenyi Biotec). Isolated CD4+ T cells were plated at a density of 0.8 × 10⁶/ml in AIM-V media (#12055091, ThermoFisher) supplemented with CTS Immune Cell SR (#A2596101 Gibco, ThermoFisher), activated with anti-CD3/anti-CD28 activation beads per manufacturer's instructions (#11131D, Gibco Dynabeads Human T-Activator CD3/CD28 for T Cell Expansion and Activation, ThermoFisher), stimulated with IL-2 (#202-IL-050, R&D), and/or T cell differentiating cytokines and antibodies (Th1: anti-IL-4 (#130-095-753), IL-12 (#130-096-704) (Miltenyi Biotec), Treg: TGFβ (#11343160, Immunotools), Th17: TGFβ, IL-1β (#201-LB-005, R&D), IL-23 (#1290-IL-010, R&D), IL-6 (#206-IL-010, R&D), anti-IFNγ (#130-095-743, Miltenyi Biotec)) +/- activated PC (Cambridge Bioscience). Where stated, T cells were incubated with +/- 20 nM activated PC (#HCAPC-0080, Cambridge Bioscience), activated PC^ΔGLA (#10175, Cambridge ProteinWorks) or activated PC^DEGR (#HCAPC-DEGR, Cambridge

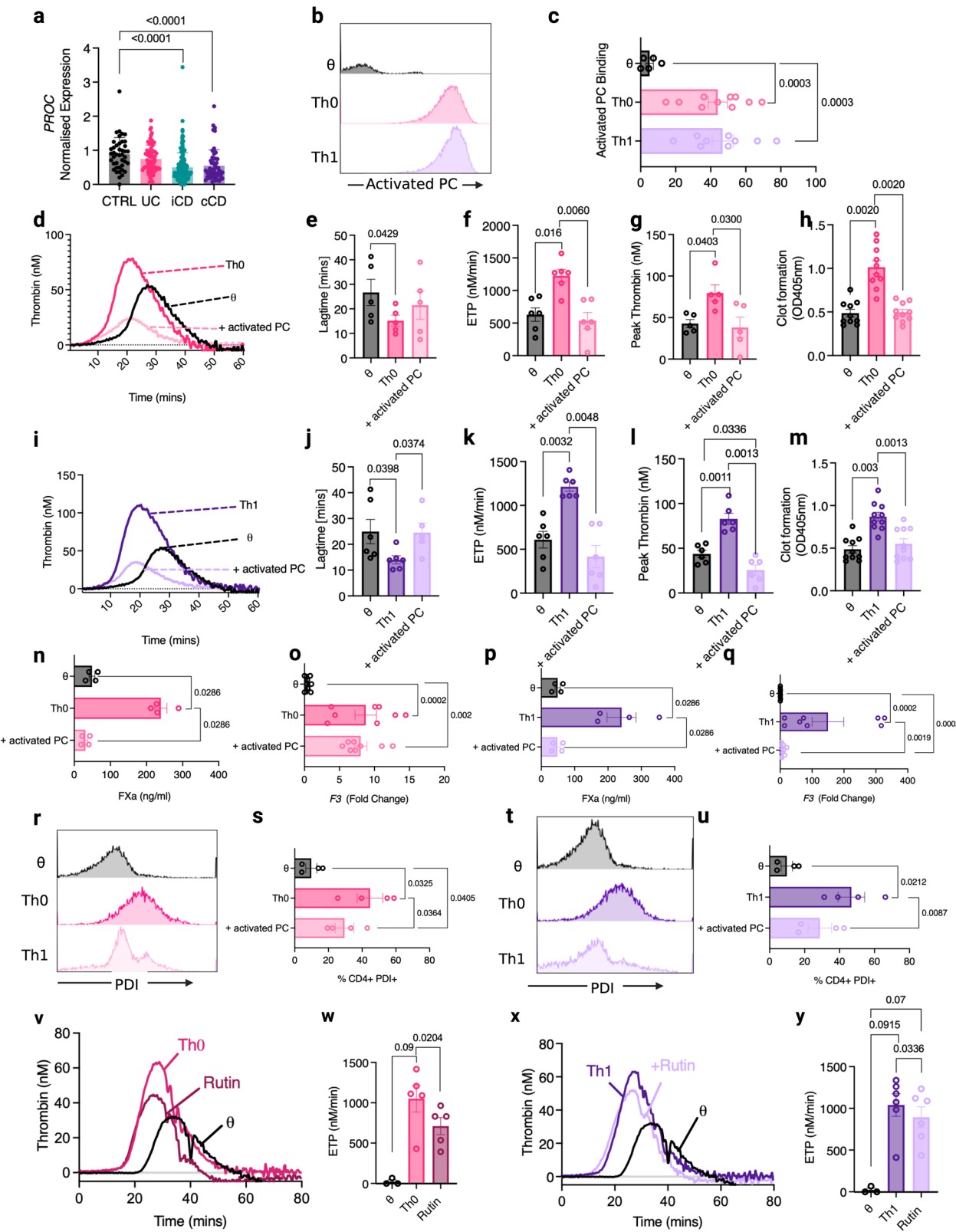

Bioscience) at 37 °C for 5 days. Isolated IBD patient and non-IBD control CD4$^+$ T cells were plated at a density of $1 \times 10^6$/ml in AIM-V media supplemented with CTS Immune Cell SR.

## T cell transfer mediated colitis model
T effector cells (CD4$^+$ CD25$^-$ CD45RB$^{hi}$) were FACs sorted from male C57BL/6 mice (#000664, Jackson laboratories) and injected i.p. into

$Rag1^{-/-}$ ($n = 4$ male) (#002216, Jackson laboratories) recipient mice ($5 \times 10^5$). To mitigate potential confounding effects of microbiota cohoused littermate control $Rag1^{-/-}$ recipient mice were injected i.p. with PBS ($n = 3$ male). Disease progression was measured by percentage weight loss compared to the original weight, with a 20% loss of original weight at the cut-off point. Colons were harvested for analysis following cervical dislocation when clinical signs of colitis

**Fig. 6 | Activated PC is downregulated in IBD and limits thrombogenic CD4+ T cell activity. a** *PROC* (Protein C; PC) expression in colonic biopsies from patients in the RISK study (GEO ID GSE57945) (iCD, *n* = 162, cCD, *n* = 56, UC, *n* = 62, control non-IBD group, *n* = 42). **b, c** Following cell culture, T cells were washed and incubated with fluorescently labelled activated PC. Binding was measured by flow cytometry. Following activated PC pre-treatment, activated PC was removed, θ (unactivated cells) and Th0 cells were washed with EDTA-containing PBS, and their capacity to initiate clotting was analysed by **d–g** thrombin generation and **h** clot formation assays. Activated PC pre-treated Th1 cell-dependent thrombin generation was analysed by **i–l** thrombin generation assays and **m** clot formation assays. Following activated PC pre-treatment, Th0 procoagulant activity was analysed by **n** FXa generation assay, **o** *F3* gene expression and **r, s** PDI cell surface expression by flow cytometry. Similarly, Th1 cell thrombogenicity was analysed by **p** FXa

generation assay, **q** *F3* gene expression and **t, u** PDI cell surface expression by flow cytometry. The contribution of PDI-mediated TF decryption was assessed by treating Th0 (**v, w**) and Th1 (**x, y**) cells with 10 mM rutin for 1 h. Cells were washed with EDTA-containing PBS, and their capacity to initiate clotting was analysed by **v–y** thrombin generation. Student's *t*-test (two-tailed) (**c**), Mann–Whitney U Test (two-tailed) (**a, n–q**), Student's paired *t*-test (two-tailed) (**e–g, j–l, m, s, u, w, y**), or Wilcoxon test (**h**) was used to determine statistical significance. Data is expressed as mean ± s.e.m. (**a, c, e–h, j–l, m–q, s, u, w, y**) for **a** 322 biological donors (iCD, *n* = 162, cCD, *n* = 56, UC, *n* = 62, control non-IBD group, *n* = 42), **c** 5–10 biological donors (*n* = 5 θ, *n* = 10 Th0 and *n* = 9 Th1), **e–g, j–l** 6 biological donors, **h, m** 10 biological donors, **n, p, s, u** 4 biological donors, **o, q** 8 biological donors and **w, y** 3–5 biological donors (*n* = 3 θ, *n* = 5 Th0/Rutin and *n* = 5 Th1/Rutin). Source data are provided in the Source Data file.

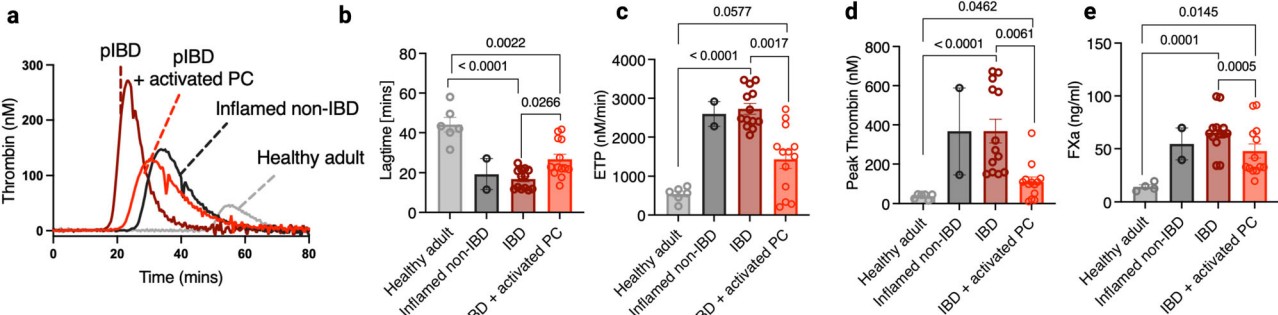

**Fig. 7 | Activated PC inhibits TF-mediated T cell thrombogenicity in IBD patient peripheral CD4+ T cells.** CD4+ T cells were isolated from donor adult human peripheral blood, inflamed non-IBD paediatric peripheral blood and paediatric IBD peripheral blood, plated at a density of 1 × 10⁶/ml and incubated at 37 °C in AIM-V media supplemented with immune replacement serum, +/- 20 nM of activated PC for 24–48 h. **a–d** Cells were washed in EDTA-containing PBS, and their ability to initiate thrombin generation was analysed by calibrated automated thrombinography in FXII-deficient plasma. **b** Lagtime, **c** ETP and **d** peak thrombin levels were measured and compared, as was their ability to facilitate FXa generation (**e**).

Mann–Whitney U Test (two-tailed) (**b–d**) or Student's *t*-test (two-tailed) (**e**) was used to determine statistical significance for all conditions except IBD vs IBD + activated PC. For these conditions, either the Wilcoxon test (two-tailed) (**b–d**) or the Student's paired *t*-test (two-tailed) (**e**) was used as the data is matched. Data is expressed as mean ± s.e.m. for **b–d** 21 biological donors (*n* = 13 IBD/IBD activated PC, *n* = 2 inflamed non-IBD, *N* = 6 healthy adult) and **e** 19 biological donors (*n* = 13 IBD, *n* = 2 inflamed non-IBD, *N* = 4 healthy adult). Source data are provided in the Source Data file.

were evident (4 weeks post transfer), and histology was performed to confirm colitis. Harvested colon tissue was fixed in 10% formalin (#HT501128-4L, Sigma Aldrich, Merck) overnight, dehydrated and embedded into paraffin blocks. Paraffin blocks were sectioned to 5 μm thickness using a microtome, mounted on Superfrost Plus adhesion slides (#1014356190, ThermoFisher), and stained with haematoxylin (#51275, Sigma Aldrich) and eosin (#1098441000, Sigma Aldrich). Histological disease scoring was performed blinded semiquantitatively from 0 to 5, as described[55]. When no changes were observed sections were graded 0. Minimal scattered immune cell infiltrate to the mucosa, with or without minimal epithelial hyperplasia was graded 1; scattered to diffuse mild immune cell infiltrates to the mucosa, sometimes extending into the submucosa, and associated with erosions, with minimal to mild epithelial hyperplasia, and minimal to mild goblet cell mucin depletion was graded 2; grade 3 encompassed mild to moderate immune cell infiltrates that were sometimes transmural, often associated with ulceration, with moderate epithelial hyperplasia and mucin depletion; marked immune cell infiltrates that were often transmural and associated with ulceration, marked epithelial hyperplasia and mucin depletion was graded 4; marked transmural immune cell infiltrates, severe ulceration and loss of intestinal glands was graded 5.

All mice were housed under specific pathogen-free conditions, on a 12-h light/dark cycle, in a temperature-controlled unit at the Comparative Medicine Unit in Trinity Translational Medicine Institute, St. James Hospital, Dublin, Ireland. Food and water were provided ad libitum. The Health Products Regulatory Authority approved all animal experiments under Project License AE19136/P125, with approval by the institutional (Trinity College Dublin) ethical review boards.

### CD4+ T cell-based thrombin generation assay
Following T cell culture, supernatants were removed, and cells were washed three times with PBS containing EDTA (#E8008, Sigma Aldrich). 20 μl MP-reagent (#86222, Stago, Fannin) and 80 μl of human normal pooled platelet-poor plasma (NPP; #CCN-15, Fanin), factor XII (FXII)-deficient plasma (#FXII-ID-50, Prolytix, Cambridge Bioscience), or factor VII (FVII)-deficient plasma (#FVII-ID-50, Prolytix, Cambridge Bioscience) were added to the washed T cells. The assay was then initiated with 20 μl of FluCa (#86197, Stago, Fannin), and fluorescence was recorded using Thrombinoscope software.

### CD4+ T cell-based clot formation assay
Following T cell culture, cell supernatants were removed, and cells were washed three times with PBS containing EDTA. 80 μl of normal pooled plasma (38%), TBS-T (52%) (#524750, Sigma Aldrich) and phospholipids (16 μM;10%) (#PL604T, Rossix, Quadratech) were added to the cells. The assay was then initiated with 20 μl of $CaCl_2$ (10.6 mM; #C8106, Sigma Aldrich) and was read at $OD_{405}$ and recorded in a kinetic assay every minute for 1 h using a Synergy MX microplate reader (BioTek).

### Factor Xa generation assay
T cell surface TF activity was measured by T cell-mediated factor Xa (FXa) generation. Following T cell culture, cell supernatants were removed, and cells were washed three times with Buffer A (10 mM Hepes (#H0887, Sigma Aldrich), 0.15 M NaCl (#71386, Sigma Aldrich), 4 mM KCl (#60142, Sigma Aldrich), 11 mM glucose (#49163, Sigma Aldrich), pH 7.5). Cells were then incubated at 37 °C with 200 μl of Buffer B (Buffer A, 5 mM $CaCl_2$, 1 mg/ml BSA (#A2153, Sigma Aldrich)

containing FVIIa (10 nM; #HCVIIA-0031, Prolytix, Cambridge Bioscience) and FX (175 nM; #HCX-0050, Cambridge Bioscience)) for 30 min. 25 μl of the supernatant was removed and added to 50 μl TBS:BSA buffer (50 mM Tris-HCL, 0.15 mol/L NaCl, 1 mg/ml BSA, 25 mol/L EDTA, pH 7.5). 50 ml of this mixture was transferred to a 96-well plate, and 50 μl of Factor Xa Chromogenic Substrate (BIOPHEN #CS-11(65), Quadratech) was added. The rate of colour development was measured against known FXa concentrations (#HCXA-0060, Cambridge Bioscience) using a Synergy MX microplate reader (BioTek).

## ELISAs
ELISA kits for human IFNγ (#88731688), TNFα (#88-7346-88), IL-17a (#88-7176-86) and IL-10 (#88-7106-86) were purchased from Invitrogen (ThermoFisher) and performed according to the manufacturer's instructions using Corning High Binding ELISA plates (#CLS336, Merck). All ELISAs were analysed using a Synergy MX microplate reader (BioTek).

## mRNA isolation and RT-qPCR of CD4$^+$ T cells
Following T cell culture, supernatants were removed, and cells were washed once with PBS and transferred to 1.5 ml RNAse free Microfuge Tubes (#AM12400, Invitrogen, ThermoFisher). Cells were then lysed using 350 μl lysis buffer, and total RNA was isolated according to the manufacturer's instructions (#12183025, PureLink RNA isolation kit, ThermoFisher). Gene expression was determined by RT-qPCR, in duplicate with Power Up SYBR green master mix (#A25742, Thermo-Fisher) using a 7500 Fast system (Applied Biosystems). Relative-fold changes in mRNA expression were calculated using the cycle threshold ($C_T$) and normalised to the *RPS18* housekeeping gene. Primer sequences are *F3* (Sense: 5′ CAGAGTTCACACCTTACCTGGAG 3′; Anti-sense: 5′ GTTGTTCCTTCTGACTAAAGTCCG 3′) & RSP18 (Sense: 5′ GCAGAATCCACGCCAGTACAAG 3′; Antisense: 5′ GCAGAATCCACGCC AGTACAAG 3′).

## Flow cytometry
All fluorescently labelled antibodies were purchased from Thermo-Fisher (αCD4, αCD3, αIFNg, αIL-17a, αFOXP3 (5 μl/test)). Cell viability was measured using the LIVE/DEAD Fixable Dead Cell Stain Kit (Aqua, Scarlet & Near IR) (Invitrogen, ThermoFisher). Before staining, cells were incubated with an anti-CD16/CD32 monoclonal antibody to block Fc receptors (20 μl/test) (Invitrogen). Cells were washed using PBS and incubated with LIVE/DEAD dye (1:100) for 30 min at 4 °C. Cells were washed using PBS supplemented with 2% FBS (#A5670801, Gibco, ThermoFisher) and then incubated with fluorescently labelled antibodies for 1 h at 4 °C (5 μl/test). For unconjugated antibodies (αASM (2 μg/test) (Invitrogen) and αPDI (1 μg/test) (Invitrogen) & αTF (2.5 μg/test) (R&D)), cells were incubated for 1 h at room temperature with the primary antibody, washed and incubated with a specific secondary antibody (1:500) for 1 h at 4 °C (Donkey anti-Goat IgG (H+L) Cross-Adsorbed Secondary Antibody Alexa Fluor 555 and Invitrogen Goat anti-Rabbit IgG (H+L) Highly Cross-Adsorbed Secondary Antibody Alexa Fluor 555, ThermoFisher). Fluorescence Minus One (FMO) and/or isotype controls were used to assess positive staining. Intracellular protein expression was evaluated by first restimulating the cells with phorbol-12-myristate 13-acetate (PMA; 10 ng/ml) (#P1585, Sigma Aldrich), Ionomycin (1 μg/ml) (#I0634, Sigma Aldrich) and Brefeldin A (5 μg/ml) (#00-4506-51, ThermoFisher) for 4–6 h at 37 °C. A FOXP3 staining buffer set (#00-5523-00, ThermoFisher) was used in accordance with the manufacturer's instructions to fix and permeabilise cells after surface staining to facilitate the detection of intracellular cytokines. Intracellular fluorescently labelled antibodies were incubated for 1 h at 4 °C. Multi-parameter analysis was then performed on an LSR/Fortessa (BD) or Attune Nxt (ThermoFisher) and analysed using FlowJo 10 software. Gating strategies for each

experiment can be found in Supplementary Fig. 8. Antibodies and dyes are listed in Supplementary Table 1.

## Cell binding assays
Recombinant human lactadherin (#2767-MF-050, R&D Systems) and human activated PC (Cambridge Bioscience) were fluorescently labelled using Lightning-Link Rapid Alexa Fluor 488 Antibody Labelling Kits in accordance with the manufacturer's instructions (#332-0030, Novus Biologicals, R&D). Before staining, cells were first incubated with an anti-CD16/CD32 Monoclonal Antibody to block FC receptors (20 μl/test) (Invitrogen, ThermoFisher). Cells were washed using PBS and incubated with LIVE/DEAD dye (1:100) for 30 min at 4 °C. Cells were washed using PBS supplemented with 2% FBS and incubated with CD4 antibodies for 30 min at 4 °C. Cells were washed and incubated with the fluorescently labelled proteins for 2 h. Fluorescence Minus One (FMO) and/or isoclonic controls were used to assess positive staining. Multi-parameter analysis was then performed on an LSR/Fortessa (BD) or Attune Nxt (ThermoFisher) and analysed using FlowJo 10 software.

## Immunofluorescence
Paraffin-embedded blocks of colon biopsies from children diagnosed with IBD ($n = 4$) or healthy controls ($n = 3$) were obtained from the DOCHAS study at the gastroenterology unit at Children's Health Ireland (CHI), Crumlin (Dublin, Ireland). Blocks were sectioned to 5 μm thickness using a microtome and mounted on Superfrost Plus adhesion slides (ThermoFisher). Antigen retrieval was performed using IHC Antigen Retrieval Solution (#00-4955-58, eBioscience, ThermoFisher) in a microwave. Tissue was probed with α-human CD3 (10 μg/ml) (eBioscience, ThermoFfisher), α-human CD4 (10 μg/ml) (eBioscience, ThermoFisher), α-human F4/80 (5 μg/ml), α-human CD11b (5 μg/ml), α-human TF (2.5 μg/ml) (R&D), or isotype controls. Paraffin-embedded blocks of colons from *Rag1*$^{-/-}$ mice adoptively transferred with wild-type T effector cells, or PBS, were sectioned to 5 μm thickness using a microtome and mounted on Superfrost Plus adhesion slides (ThermoFisher). Antigen retrieval was performed using IHC Antigen Retrieval Solution (eBioscience, ThermoFisher) in a microwave. Tissue was probed with α-mouse CD3 (1:100) (FITC) (eBioscience, Thermo-Fisher), α-mouse TF (2.5 μg/ml) (R&D) or isotype controls. The secondary antibodies used were donkey anti-Goat IgG (H+L) Cross-Adsorbed Secondary Antibody Alexa Fluor 555 (1 μg/ml) and Invitrogen Goat anti-Rabbit IgG (H+L) Highly Cross-Adsorbed Secondary Antibody Alexa Fluor 555 (1 μg/ml). Tissue was mounted using SlowFade Gold antifade Mountant with DAPI (#S36942, ThermoFisher). Images were taken using a confocal microscope Zeiss LSM700. Positive cells stained with αCD3, αCD4, αF480, αCD11b and αTF were quantified using the ImageJ tool Fiji and the Count plugin. An average of three different images per sample were analysed statistically using GraphPad Prism 9.5 software to give an average number of cells/field. Antibodies and dyes are listed in Supplementary Table 1.

## RNAseq data analysis
We utilised multi-omic data from publicly available datasets of IBD patient intestinal biopsies, including RNA-seq data from the Risk Stratification and Identification of Immunogenetic and Microbial Markers of Rapid Disease Progression in Children with Crohn's Disease (RISK) study, in which we compared the expression of genes of interest between paediatric CD and UC patients and healthy controls. We accessed NCBI Gene Expression Omnibus datasets using GEO ID GSE57945 (Risk Cohort). The log2 fold-change and p-value significance data were downloaded and analysed using GraphPad Prism 9.5 software. Data are shown as means ± SEM.

We also utilised in-house RNA-seq data generated from paediatric IBD patients and control participants' biopsies recruited in the DOC-HAS study at the Children's Health Ireland (CHI) gastroenterology unit

(GEO ID GSE266325). Total RNA was prepared from patient-derived colonic punch biopsies within 2 h of harvesting using Isolate II RNA Mini Kit (Bioline). A total of 300 ng of total RNA per patient sample was shipped on dry ice for RNA sequencing using the Illumina Novoseq 6000 Sequencing Platform by Novogene (Cambridge, USA). Briefly, total RNA purity and integrity were confirmed prior to poly-A capture and mRNA enrichment for reverse transcription to cDNA. A 150 bp Paired End strategy was employed for sequencing. Raw data was filtered using fastp software to remove low-quality reads prior to downstream analysis. Paired-end clean reads were aligned to the reference genome using Hisat2 v2.0.5. For the quantification of gene expression levels featureCounts v1.5.0-p3 was used and Fragments Per Kilobase of transcript per Millions base pairs sequenced (FPKM) was calculated. Relative expression levels for selected genes of interest between patient groupings were analysed based on mean FPKM values for each group. Relative gene expression per individual gene is shown as z scores in a heatmap. KEGG pathway analysis was used to depict dysregulation in the coagulation pathway.

### Isolation of colonic lamina propria cells

Paediatric IBD and non-IBD biopsies were obtained from the DOCHAS study, National Centre for Paediatric Gastroenterology, CHI-Crumlin, Dublin Ireland. Lamina propria cells were isolated from these biopsies using a modified version of the Smillie et al. Star Method[56]. Four biopsies per patient were collected in AIM media (ThermoFisher) supplemented with CTS Immune Cell SR (Gibco, ThermoFisher). Biopsies were first rinsed in 30 ml of ice-cold PBS (Sigma Aldrich, Merck) and then transferred to 10 ml epithelial cell solution (HBSS Ca/Mg-Free (#14170112, ThermoFisher), 10 mM EDTA (Sigma Aldrich, Merck), 100 U/ml penicillin & 100 mg/ml streptomycin (#15070063, ThermoFisher), 10 mM HEPES (Sigma Aldrich, Merck), and 2% FCS (ThermoFisher) freshly supplemented with 200 µl of 0.5 M EDTA). The epithelial layer was separated from the underlying lamina propria by shaking horizontally at 37 °C for 15 min at 200 rpm. The tube was then placed on ice for 10 min and shaken vigorously 15 times. The tissues were carefully removed and placed into 10 ml of ice-cold PBS to rinse and transferred to 5 ml of enzymatic digestion mix (RPMI1640 (#21875034, ThermoFisher), 100 U/ml penicillin, 100 mg/ml streptomycin, 10 mM HEPES, 2% FCS, 50 mg/ml gentamicin (#15710064, ThermoFisher), 100 mg/ml of Liberase (#LIBTM-RO, Roche, Merck) and 100 mg/ml of DNase I (#D4263, Sigma Aldrich, Merck)). The tubes were shaken horizontally at 37 °C for 30 min at 200 rpm. After 30 min, 1 ml of FCS and 80 µl of 0.5 M EDTA were added to quench the digestion mix and the tubes were placed on ice for 5 min. The dissociated lamina propria cell solution was then filtered through a 40-µm cell strainer into a new 50 ml conical tube and rinsed through with PBS. The tube was then spun down at 400×g for 10 min, and the pelleted live cells were counted using Trypan blue exclusion (#15250061, ThermoFisher). Cells were resuspended at $1 \times 10^6$/ml in AIM-V media supplemented with CTS Immune Cell SR (Gibco, ThermoFisher), and incubated at 37 °C for 24–48 h prior to analyses by flow cytometry.

### Statistical analysis

For each dataset, statistical analysis was performed by first analysing the data for normal distribution (Shapiro–Wilk or Kolmogorov–Smirnov test) and equality of variance (F test). Unpaired data was then analysed by two-tailed Student's t-test or one-way ANOVA for parametric data or Mann–Whitney U test for nonparametric data, as appropriate. Paired data was analysed by paired two-tailed Student's t-test for parametric data, or Wilcoxon matched-pairs signed ranks test for nonparametric data, as appropriate. All analysis and graph representation were performed using GraphPad Prism 9.5 software. Data are shown as means ± s.e.m. or means ± s.d.

### Reporting summary

Further information on research design is available in the Nature Portfolio Reporting Summary linked to this article.

### Data availability

All data included in the Supplementary Information are available from the authors, as are any unique reagents used in this article. The raw numbers for charts and graphs are available in the Source Data file whenever possible. The RNA-seq data analysed in this study from paediatric IBD patients is available from the NCBI Gene Expression Omnibus database under accession code GSE266325. RNA-seq data analysed in this study from IBD patients in the RISK cohort is available from the NCBI Gene Expression Omnibus database under accession code GSE57945. Source data are provided with this paper.

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

## Acknowledgements

The authors would like to thank Dr Ingmar Schoen for his contribution to the analysis of immunofluorescent data. Grant support for R.J.S.P. is provided by Science Foundation Ireland (21/FFP-A/8859), The National Children's Research Centre (C/18/3) and Health Research Board (ILP-POR-2022-060). Grant support for P.T.W. is provided by Science Foundation Ireland (21/FFP-P/10135).

## Author contributions

G.L. and R.J.S.P. devised the study and experimental strategy and wrote the manuscript. P.A.K., A.M.R., G.L., S.H., S.C.B., J.S.O.'D., and P.T.W. performed experiments and analysed data. S.H., S.E.J.C. and A.D. recruited and consented patients, collected samples and patient information. All authors reviewed and contributed to the final version of the manuscript.

## Competing interests

The authors declare no competing interests.
