## [Transparent Peer Review file · Nature Communications]

Tissue factor-dependent colitogenic CD4+ T cell thrombogenicity is regulated by activated protein C signaling.

Corresponding Author: Dr Roger Preston

Version 0:

Reviewer comments:

Reviewer #1

(Remarks to the Author)

This very well written paper significantly advances mechanistic insights related to procoagulant and anticoagulant mechanisms relevant to inflammatory bowel disease (IBD), as summarized in the Abstract. The paper identified tissue factor (TF)-positive CD4+CD3+ T cells in the colons of colitogenic mice and in paediatric IBD patients and showed that inflammatory CD4+ T cells express TF in its procoagulant activated ("decrypted") state. The paper provides functional studies showing that activated protein C (APC) signalling reduced colitogenic T cell generation and activity, potentially impaired TF decryption, and significantly reduced T cell-mediated thrombin generation and clot formation. In summary, the data identify TF-mediated colitogenic T cell thrombogenicity and demonstrate a key role for APC signalling in regulating T cell thrombo-inflammatory activity. Consequently, these findings highlight a potential therapeutic role for APC to target CD4+ T cell thrombo-inflammatory activity in IBD and other T cell-mediated disease contexts.

Major Comments for Revisions

None.

Minor Comments for Revisions

1. page 9, line 16.

The text should be modified as follows. The text stating "... therefore excluded a role for FXII activation in the observed cell-dependent thrombin generation..." should be replaced with text saying "...therefore showing a requirement for factor VII in the observed cell-dependent thrombin generation..."

2. page 14, lines 23-26.

The authors wisely interpret their paper to point towards potential novel therapy for IBD and for regulating T cell-specific immunothrombotic activity using recombinant APC non-anticoagulant variants. But some references should be provided here describing such variants to enable readers to get some reference(s) about this highly relevant topic and to appreciate this suggestion for translational development. Suggestions -- References here should include the authors' 2022 review, reference 49 (although this ref.49 is not in conveniently accessible (fee required)). So referencing here might also include additional references for readers who are not familiar with the APC/protein C literature. Other recent review(s) about non-anticoagulant APC could be cited to enrich the text. E.g., the authors might cite their own review, reference 33 (which is free to download), which cites papers about non-anticoagulant APC.

3. page 3, line 4.

The introductory referencing related to IBD and the pathogenic roles for certain T cells could be enhanced by adding a reference to the recent insightful paper about a role for stem-like CD4+ and CD8+ T cells in the pathogenesis of ulcerative colitis by Y. Li et al, P. Vijavanand. Nature Immunology 2024.

Reviewer #2

(Remarks to the Author)

The work of Leon et al investigates an interesting topic, ie the evaluation of T cell-dependent thrombogenicity in intestinal inflammation.

Hypothesis of the work is to understand the cellular mechanisms leading to an increased risk of venous thromboembolisms in patients suffering from inflammatory bowel disease.

Authors identify CD4+ T cells as a relevant cellular source of decrypted TF, leading to thrombin generation. Additionally, dysregulated expression of the protein C pathway contributes to T-cell mediated thrombogenicity. Indeed, a functional APC signaling reduces T cell-mediated thrombin generation and clot formation.

The work is interesting, easy to follow and addressing a relevant clinical problem whose mechanistic bases are poorly understood. However, some points require additional clarifications to strengthen the message of the study.

Major points:

1. Authors focus their attention on CD4+T cells, utilizing Th0 e Th1 cells as proxy to demonstrate T cells' involvement in this functional pathway and dissect the components. However, as they clearly showed, Th17 cells are also modulated but they are not tested during the functional in vitro and in vivo assays. Same can be said for Tregs (additionally: why Th1 and Th17 cells were analysed also through cytokines' secretion while treg only with FoxP3 expression?)

Since Th1/Th17 cells are driving intestinal inflammation in CD patients, it should be highly informative to perform experiments also with this T cell subset.

Also, it could be important also to evaluate TNF production in addition to IFN γ .

2. Figure 2F: who are the other TF+CD3 neg cells? They seem quite abundant and may be extremely relevant in this context.

3. Figure 4 (and previous): authors should provide a gating strategy of T cells

4. Although it could be important to validate the modulation of T cells thrombogenicity with freshly isolated T cells from IBD patients, and might not be possible to obtain them, authors should show at least that in the murine models that APC-pretreatment limits or controls T cell thrombogenicity and TF production in vivo.

Minor points

_Statistics in supp5 needs to be thoroughly revised. I hardly believe in the stat of panels d and h.

_fig 3K: I do not understand the meaning of putting a histogram without reference of frequency or MFI within the samples. Fig

3I: this is not TF surface expression but the frequency of CD4+TF+ cells

Reviewer #3

(Remarks to the Author)

The manuscript by Leon et al is focused on a series of studies that evaluate the expression of tissue factor by T cells in clinical and mouse models of IBD. The results suggest that CD4 T cells are among the TF expressing cell types in the diseased colon and that upregulation of TF expression/activity is evident in activated T cells in vitro. Mechanisms controlling TF decryption were evaluated and similarly increased. Finally, the results suggest the possibility that APC plays a key signaling role (non-anticoagulant function) in regulating TF expression by CD4 T cells, with substantial disease implications. Definitely an interesting manuscript with several novel findings. I do have several comments worth considering.

1) The rationale for the study is framed on increased risk and complications from VTE in patients as well as potential disease contributing function of TF. Although the results document expression of TF by CD4 T cells and an impact of APC treatment on these cells, the precise role of TF expressed by T cells in either pathology is unexplored. Pursuing studies using TFflox/flox mice to document the role of T cell TF in the experimental model would strengthen the manuscript. This comment acknowledges that the core novel findings are that TF is expressed by these cells in this context.

2) The results include multiple elements indicating expression of TF by T cells, from immunolabeling to activity detected by thrombin generation or two-stage coagulation assay ex vivo. These are strong results. Figure 4 suggests decryption of TF procoagulant activity, but the results provided are biomarkers of processes contributing to TF decryption, not decryption itself. Do interventions blocking PDI, PS etc. reduce TF activity to levels observed in cells? What is the relative contribution of TF mRNA induction and decryption?

3) Details on duration of pre-treatment with APC should be included in Figure 5. Moreover, the precise mechanism whereby APC is mediating inhibition of TF expression and encryption is unclear, although multiple possibilities are mentioned in the discussion. This feels important to build on with experiments in the manuscript, as there is limited evidence from in vivo studies that changes in local or plasma APC are changing TF expression. Are the PARs involved in this context, or EPCR?

Version 1:

Reviewer comments:

Reviewer #2

(Remarks to the Author)

Authors performed a thorough revision of the manuscript, answering satisfactorily to all the queries.

This reviewer appreciate the great effort demonstrated, including the successful completion of the validation experiments with the human IBD samples.

I have nothing else to add

Reviewer #3

(Remarks to the Author)

I appreciate the authors thoughtful response to my comments, including addition of new experiments. The results provided a framework on which follow-on in vivo experimentation can be performed. Nice paper.

REVIEWER COMMENTS

Reviewer #1 (Thrombosis) (Remarks to the Author):

This very well written paper significantly advances mechanistic insights related to procoagulant and anticoagulant mechanisms relevant to inflammatory bowel disease (IBD), as summarized in the Abstract. The paper identified tissue factor (TF)-positive CD4+CD3+ T cells in the colons of colitogenic mice and in paediatric IBD patients and showed that inflammatory CD4+ T cells express TF in its procoagulant activated ("decrypted") state. The paper provides functional studies showing that activated protein C (APC) signalling reduced colitogenic T cell generation and activity, potently impaired TF decryption, and significantly reduced T cell-mediated thrombin generation and clot formation. In summary, the data identify TF-mediated colitogenic T cell thrombogenicity and demonstrate a key role for APC signalling in regulating T cell thrombo-inflammatory activity. Consequently, these findings highlight a potential therapeutic role for APC to target CD4+ T cell thrombo-inflammatory activity in IBD and other T cell-mediated disease contexts.

Major Comments for Revisions

None.

Minor Comments for Revisions

1. page 9, line 16.

The text should be modified as follows. The text stating "... therefore excluded a role for FXII activation in the observed cell-dependent thrombin generation..." should be replaced with text saying "...therefore showing a requirement for factor VII in the observed cell-dependent thrombin generation..."

This sentence has been modified as suggested.

2. page 14, lines 23-26.

The authors wisely interpret their paper to point towards potential novel therapy for IBD and for regulating T cell-specific immunothrombotic activity using recombinant APC non-anticoagulant variants. But some references should be provided here describing such variants to enable readers to get some reference(s) about this highly relevant topic and to appreciate this suggestion for translational development. Suggestions -- References here should include the authors' 2022 review, reference 49 (although this ref.49 is not in conveniently accessible (fee required)). So referencing here might also include additional references for readers who are not familiar with the APC/protein C literature. Other recent review(s) about non-anticoagulant APC could be cited to enrich the text. E.g., the authors might cite their own review, reference 33 (which is free to download), which cites papers about non-anticoagulant APC.

We agree that additional referencing to primary/review articles relating to APC non-anticoagulant variants would be valuable, and have cited these in the revised manuscript.

3. page 3, line 4.

The introductory referencing related to IBD and the pathogenic roles for certain T cells could be enhanced by adding a reference to the recent insightful paper about a role for stem-like CD4+ and CD8+ T cells in the pathogenesis of ulcerative colitis by Y. Li et al, P. Vijavanand. Nature Immunology 2024.

We have added this citation to the revised manuscript.

Reviewer #2 (IBD, mucosal immunity) (Remarks to the Author):

The work of Leon et al investigates an interesting topic, ie the evaluation of T cell-dependent thrombogenicity in intestinal inflammation.

Hypothesis of the work is to understand the cellular mechanisms leading to an increased risk of venous thromboembolisms in patients suffering from inflammatory bowel disease.

Authors identify CD4+ T cells as a relevant cellular source of decrypted TF, leading to thrombin generation. Additionally, dysregulated expression of the protein C pathway contributes to T-cell mediated thrombogenicity. Indeed, a functional APC signaling reduces T cell-mediated thrombin generation and clot formation.

The work is interesting, easy to follow and addressing a relevant clinical problem whose mechanistic bases are poorly understood. However, some points require additional clarifications to strengthen the message of the study.

Major points:

1. Authors focus their attention on CD4+T cells, utilizing Th0 e Th1 cells as proxy to demonstrate T cells' involvement in this functional pathway and dissect the components. However, as they clearly showed, Th17 cells are also modulated but they are not tested during the functional in vitro and in vivo assays. Same can be said for Tregs (additionally: why Th1 and Th17 cells were analysed also through cytokines' secretion while treg only with FoxP3 expression?) Since Th1/Th17 cells are driving intestinal inflammation in CD patients, it should be highly informative to perform experiments also with this T cell subset.

We agree that evaluation of the thrombogenic potential of alternative T cell subsets would add valuable information to the revised manuscript. To achieve this, we generated Th17 cells, treated them with APC and assessed their ability to support TF-dependent thrombin generation in plasma. We observed that APC slightly reduces the number of CD4+IL17A+ cells, albeit not significantly (**Supplementary Figure 6e-g**). As with Th0/Th1 cells, Th17 cells supported thrombin generation more effectively in naïve T cells, as demonstrated by an accelerated time to thrombin generation and enhanced thrombin generation parameters (**Supplementary Figure 6h-k**). Similarly, FXa generation on the surface of Th17 cells was approximately five-fold greater than on naïve T cells (**Supplementary Figure 6l**). Like Th1 cells, the thrombogenic activity of Th17 cells was attenuated by pre-treatment with APC, which increased lag-time and reduced FXa generation (**Supplementary Figure 6h-l**). These new data indicate that Th17 cells, like Th0/Th1, possess the capacity to initiate blood clotting, and this activity can be suppressed by APC signalling.

Next, we generated inducible Tregs to assess thrombogenic activity. Notably, treatment with APC increased the number of CD4+FOXP3+ T cells (**Supplementary Figure 6m-o**), in keeping with previous reports^{1,2}. We also observed increased TF-dependent FXa and plasma thrombin generation in the presence of iTregs, and this activity could also be inhibited by pre-treatment with APC (**Supplementary Figure 6p-t**). However, iTreg populations typically consist of 20-40% Th0 cells³⁴, which, as we have described, also possess significant thrombogenic potential. As it is highly challenging to generate a sufficient number of pure CD4+FOXP3+ T cells for thrombin generation analysis, the TF-dependent procoagulant activity of iTregs may arise, at least partly, from Th0 cells that are also likely present during analysis.

Also, it could be important also to evaluate TNF production in addition to IFNg.

We also assessed TNF production from Th1 cells both in the presence and absence of APC. As expected, we observed significantly increased TNF production from Th1 cells, which was

almost entirely ablated by pre-treatment with APC, in keeping with previous studies demonstrating the anti-inflammatory activity of APC. This data has been added to **Supplementary Figure 6d**.

2. Figure 2F: who are the other TF+CD3 neg cells? They seem quite abundant and may be extremely relevant in this context.

TF expression in the colon has been described to occur most prominently on colonic epithelial and goblet cells⁵. To assess whether TF⁺ monocytes or macrophages were also present within the colon, we stained colon tissue from mice with T cell transfer-induced colitis with antibodies to CD11b and F480 (**new Supplementary Figure 4**). We found very limited numbers of TF⁺ monocytes and macrophages within the colonic tissue sections. We expect this may be explained by the nature of the model used, in which colitis is driven by T cells in this instance.

3. Figure 4 (and previous): authors should provide a gating strategy of T cells.

The revised manuscript now provides a gating strategy for all flow cytometry analyses. This has been included in **Supplementary Figure 8**.

4. Although it could be important to validate the modulation of T cells thrombogenicity with freshly isolated T cells from IBD patients, and might not be possible to obtain them, authors should show at least that in the murine models that APC-pretreatment limits or controls T cell thrombogenicity and TF production in vivo.

This is a good suggestion, that we felt could be most effectively addressed by assessment of T cell thrombogenicity from freshly isolated T cells from IBD patients, rather than in a murine model, where wildtype APC administration would cause direct anticoagulant activity, complicating analysis of its indirect anticoagulant effect via signalling on T cells. To achieve this, we were able to attain both fresh blood and colonic biopsies from treatment naïve paediatric IBD patients. We measured TF activity, thrombogenic potential, and response to APC compared to either inflamed paediatric biopsy tissue from patients later found not to meet IBD criteria (biopsy tissue) or T cells isolated from healthy individuals (blood).

First, we isolated lamina propria cells from fresh colonic biopsies of treatment naïve paediatric IBD patients. We then measured TF expression on the CD4⁺ T cells from the colonic tissue by flow cytometry. We compared TF expression to that of 'query' IBD patients, who presented to the clinic with gastrointestinal complaints and inflammation, underwent routine diagnostic testing, but were later ruled non-IBD (**Figure 1e-h**). Although this population does not have IBD, they experienced gastro-intestinal inflammation at the time of biopsy, so do not represent a fully 'healthy' control population. However, acquiring intestinal biopsies from a non-inflamed GI paediatric population is impossible, so they represent the closest control population available. Notably, TF expression was very elevated on CD4⁺ T cells isolated from both IBD and non-IBD groups, but the IBD patient cohort exhibited consistently higher TF expression (88.8% +/- 5.7 TF⁺ cells). This data complements other data in **Figure 1**, which shows that TF is elevated on tissue-infiltrating T cells during IBD.

To directly address the reviewer's query, we also isolated CD4⁺ T cells from the peripheral blood of paediatric IBD patients and the non-IBD paediatric control population. Then, we analysed the level of TF expression, activity and thrombogenicity (**new Figure 5**). As the non-IBD paediatric control population is not a 'healthy' comparison, we also isolated CD4⁺ T cells from the peripheral blood of healthy adult donors for comparative analysis. We observed only

a very small population of CD4⁺TF⁺ T cells in healthy adult blood, but the peripheral CD4⁺ TF⁺ population was significantly increased in the IBD patient population (**Figure 5a-e**).

Next, we assessed the thrombogenic potential of CD4⁺ T cells isolated from the blood of each group. The CD4⁺ T cells from IBD patients exhibited a significantly increased capacity to initiate thrombin generation relative to T cells isolated from healthy individuals (**Figure 5f-i**). Moreover, these IBD patient cohort T cells enabled 6-fold increased FXa generation than those from healthy controls (**Figure 5j**).

Finally, to ascertain whether thrombogenic T cells isolated from the peripheral blood of IBD patients were sensitive to APC-dependent suppression of TF procoagulant activity, T cells were incubated with APC, before the cells were washed and assessed by thrombin generation analysis (**new Figure 7**). APC treatment significantly extended lag time and reduced peak and total thrombin generation in T cells from IBD and inflamed non-IBD groups (**Figure 7a-d**). Moreover, prior APC treatment also reduced FXa generation mediated by peripheral CD4⁺ T cells from IBD patients (**Figure 7e**), in keeping with the earlier description of APC-mediated TF suppression on T cells.

Collectively, this extensive new data prepared for the revised manuscript significantly expands the impact of our study by describing, for the first time, the presence of procoagulant CD4⁺TF⁺ T cells in the periphery of IBD patients, further emphasising their pathogenic potential. We are therefore grateful to the reviewer for this insightful suggestion.

Minor points

_Statistics in supp5 needs to be thoroughly revised. I hardly believe in the stat of panels d and h.

To address this, we performed additional experiments to increase the number of experimental replicates, which have been added to **Supplementary Figure 6e and m** (formerly Supplementary Figure 5). Each dot represents the average of 3 technical replicates performed for 7 biological donors. The normality of the distribution of these results was then tested using the D'Agostino & Pearson test. A Wilcoxon t-test was deemed the most appropriate test to analyse the paired data from biological replicates, as a non-parametric distribution was observed.

_fig 3K: I do not understand the meaning of putting a histogram without reference of frequency or MFI within the samples.

The histogram in **Figure 3k** is a representative image of the data depicted in **Figure 3l**. We have updated this figure to highlight this better. Furthermore, the data now references the percentage of TF⁺CD4⁺T cells and the percentage present in each group.

Fig 3l: this is not TF surface expression but the frequency of CD4+TF+ cells.

We have now included this modification as a new figure panel (**Figure 3m**), and the text has been modified to reflect these changes.

Reviewer #3 (Thrombosis) (Remarks to the Author):

The manuscript by Leon et al is focused on a series of studies that evaluate the expression of tissue factor by T cells in clinical and mouse models of IBD. The results suggest that CD4 T cells are among the TF expressing cell types in the diseased colon and that upregulation of TF expression/activity is evident in activated T cells in vitro. Mechanisms controlling TF

decryption were evaluated and similarly increased. Finally, the results suggest the possibility that APC plays a key signaling role (non-anticoagulant function) in regulating TF expression by CD4 T cells, with substantial disease implications. Definitely an interesting manuscript with several novel findings. I do have several comments worth considering.

1) The rationale for the study is framed on increased risk and complications from VTE in patients as well as potential disease contributing function of TF. Although the results document expression of TF by CD4 T cells and an impact of APC treatment on these cells, the precise role of TF expressed by T cells in either pathology is unexplored. Pursuing studies using TF^{flox/flox} mice to document the role of T cell TF in the experimental model would strengthen the manuscript. This comment acknowledges that the core novel findings are that TF is expressed by these cells in this context.

To our knowledge, no T cell-specific TF knockout mouse has been described in the scientific literature to date. We agree that creating this new mouse line would be of significant value in understanding the role of TF on T cells in this context and plan to do this in the future. However, this is not a trivial task, and generating a novel transgenic mouse line would require significant resources. Furthermore, the national regulatory body in Ireland requires a specific monitoring period for newly generated transgenic mouse lines of at least two generations before any further experimentation can be performed, and requires data from preliminary studies to assess how TF deletion in T cells would impact overall physiology (which is currently unknown), adding to the overall project cost and timeline. We would then need further ethical approval from the same regulatory body before we could even begin to repeat the T cell transfer model experiments described in the current manuscript, further extending the research timeline.

With everything going according to plan, we anticipate that the generation of this new mouse and completion of a new set of data with this mouse line would take a minimum of 12-18 months. Therefore, while we greatly appreciate the potential insights that could come from a T cell-specific TF knockout mouse and have plans to pursue this in the future, the practical and biological challenges make the establishment of such a model and execution of the proposed experiments unviable within a reasonable resubmission timeframe for this manuscript. However, to further address the role of CD4⁺ T cells in vivo, we have significantly expanded our clinical analysis of CD4⁺ T cells isolated directly from IBD patient colonic tissue and, in addition, shown that peripheral T cells isolated from IBD patient blood also express TF in an active state, reflecting the physiological importance of TF⁺ CD4⁺ T cells in IBD thromboinflammatory activity (see revised **Figures 1, 5 and 7**).

2) The results include multiple elements indicating expression of TF by T cells, from immunolabeling to activity detected by thrombin generation or two-stage coagulation assay ex vivo. These are strong results. Figure 4 suggests decryption of TF procoagulant activity, but the results provided are biomarkers of processes contributing to TF decryption, not decryption itself. Do interventions blocking PDI, PS etc. reduce TF activity to levels observed in θ cells? What is the relative contribution of TF mRNA induction and decryption?

This is an excellent suggestion. As PDI activity was increased in Th0 and Th1 cells and inhibited in the presence of APC (**Figure 6r-u**), we sought to address the reviewer's query by determining whether direct PDI inhibition would result in diminished TF-mediated procoagulant activity. To explore this, we utilised rutin, a well-characterised inhibitor of PDI activity⁶ in T cell-dependent thrombin generation analysis. Interestingly, rutin-mediated PDI inhibition on CD4⁺ T cells caused a significant reduction in ETP in both Th0 and Th1 cells. This data has been added as new panels in **Figure 6v-y**. Notably, this inhibitory effect did not restore thrombin

generation to that of naïve unstimulated T cells, suggesting that while effective, PDI inhibition alone is not sufficient to prevent TF-dependent procoagulant activity on Th0 or Th1 cells. Inhibition of PS externalisation was not possible in this experimental context as cell surface PS is also required for vitamin K-dependent coagulation factor membrane interactions, hampering the assessment of a TF-specific role for PS in regulating plasma thrombin generation. Our new data suggests a role for both PDI-mediated TF decryption in regulating colitogenic T cell procoagulant activity, but does not exclude a potential role for alternative or novel methods of TF decryption that may also be employed by activated CD4⁺ T cells to promote procoagulant TF activity. Given that the mechanistic basis of TF decryption remains poorly understood on all TF-expressing cells, how TF activity is modulated on T cells, by PDI or otherwise, is of particular interest for future investigation.

3) Details on duration of pre-treatment with APC should be included in Figure 5.

Moreover, the precise mechanism whereby APC is mediating inhibition of TF expression and encryption is unclear, although multiple possibilities are mentioned in the discussion. This feels important to build on with experiments in the manuscript, as there is limited evidence from in vivo studies that changes in local or plasma APC are changing TF expression. Are the PARs involved in this context, or EPCR?

The legend describing the new **Figure 6** (previously Figure 5) now includes details on the pre-treatment duration with APC. To address the potential role of EPCR and PARs in this activity, we utilised APC variants incapable of either EPCR binding (APC^{ΔGLA}) or PAR activation (APC^{DEGR}) in our T cell-mediated thrombin generation assays. Interestingly, the loss of PAR proteolysis did not significantly affect APC restriction of T cell-mediated procoagulant activity in Th0 or Th1 cells (**Supplementary Figure 7a-f**). In contrast, loss of EPCR did cause a significant increase in some thrombin generation parameters, although the effect varied between Th0 and Th1 cells. These results indicate that APC may utilise EPCR, but is less reliant on PAR proteolysis to reduce T cell thrombogenicity. Notably, PAR-independent APC signalling has previously been described on other immune cells, including monocytes⁷ and neutrophils⁸ and EPCR occupancy alone has been shown to be sufficient to induce anti-inflammatory PAR signalling activity by otherwise pro-inflammatory PAR1 agonists⁹. However, whether APC signals directly via apolipoprotein E receptor 2⁷, various integrins⁸, or alternative T cell-specific APC receptors to mediate these activities is currently unknown and is the subject of active investigation in our lab.

References:

1. Ranjan S, Goihl A, Kohli S, et al. Activated protein C protects from GvHD via PAR2/PAR3 signalling in regulatory T-cells. *Nat. Commun.* 2017;8(1):311.
2. Gupta D, Elwakiel A, Ranjan S, et al. Activated protein C modulates T-cell metabolism and epigenetic FOXP3 induction via α-ketoglutarate. *Blood Adv.* 2023;7(17):5055–5068.
3. Leon G, Hernandez Santana YE, Irwin N, et al. IL-36 cytokines imprint a colitogenic phenotype on CD4. *Mucosal Immunol.* 3AD;15(3):491–503.
4. Gu J, Shao Q, Zhou J, Chen Q, Lu L. Protocol for *in vitro* isolation, induction, expansion, and determination of human natural regulatory T cells and induced regulatory T cells. *STAR Protoc.* 2022;3(4):101740.
5. Queiroz KCS, van 't Veer C, van den Berg Y, et al. Tissue Factor–Dependent Chemokine Production Aggravates Experimental Colitis. *Mol. Med.* 2011;17(9–10):1119–1126.
6. Liao X, Ji P, Chi K, et al. Enhanced inhibition of protein disulfide isomerase and anti-thrombotic activity of a rutin derivative: rutin:Zn complex. *RSC Adv.* 13(17):11464–11471.
7. Activated protein C ligation of ApoER2 (LRP8) causes Dab1-dependent signaling in U937 cells.

8. Elphick GF, Sarangi PP, Hyun Y-M, et al. Recombinant human activated protein C inhibits integrin-mediated neutrophil migration. *Blood*. 2009;113(17):4078–4085.
9. Roy RV, Ardeshiryajimi A, Dinarvand P, Yang L, Rezaie AR. Occupancy of human EPCR by protein C induces β -arrestin-2 biased PAR1 signaling by both APC and thrombin. *Blood*. 2016;128(14):1884–1893.